# Soft Mixture Denoising: Beyond the Expressive Bottleneck of Diffusion Models

**Yangming Li, Boris van Breugel, Mihaela van der Schaar**
Department of Applied Mathematics and Theoretical Physics
University of Cambridge
`yl874@cam.ac.uk`

## Abstract

Because diffusion models have shown impressive performances in a number of tasks, such as image synthesis, there is a trend in recent works to prove (with certain assumptions) that these models have strong approximation capabilities. In this paper, we show that current diffusion models actually have an *expressive bottleneck* in backward denoising and some assumption made by existing theoretical guarantees is too strong. Based on this finding, we prove that diffusion models have unbounded errors in both local and global denoising. In light of our theoretical studies, we introduce *soft mixture denoising* (SMD), an expressive and efficient model for backward denoising. SMD not only permits diffusion models to well approximate any Gaussian mixture distributions in theory, but also is simple and efficient for implementation. Our experiments on multiple image datasets show that SMD significantly improves different types of diffusion models (e.g., DDPM), especially in the situation of few backward iterations.

## 1 Introduction

Diffusion models (DMs) (Sohl-Dickstein et al., 2015) have become highly popular generative models for their impressive performance in many research domains—including high-resolution image synthesis (Dhariwal & Nichol, 2021), natural language generation (Li et al., 2022), speech processing (Kong et al., 2021), and medical image analysis (Pinaya et al., 2022).

**Current strong approximator theorems.** To explain the effectiveness of diffusion models, recent work (Lee et al., 2022a;b; Chen et al., 2023) provided theoretical guarantees (with certain assumptions) to show that diffusion models can approximate a rich family of data distributions with arbitrarily small errors. For example, Chen et al. (2023) proved that the generated samples from diffusion models converge (in distribution) to the real data under ideal conditions. Since it is generally intractable to analyze the non-convex optimization of neural networks, a potential weakness of these works is that they all supposed *bounded score estimation errors*, which means the prediction errors of denoising functions (i.e., reparameterized score functions) are bounded.

**Our limited approximation theorems.** In this work, we take a first step towards the opposite direction: Instead of explaining why diffusion models are highly effective, we show that their approximation capabilities are in fact limited and the assumption of *bounded score estimation errors* (made by existing theoretical guarantees) is too strong.

In particular, we show that current diffusion models suffer from an **expressive bottleneck**—the Gaussian parameterization of backward probability $p_\theta(\mathbf{x}_{t-1} \mid \mathbf{x}_t)$ is not expressive enough to fit the (possibly multimodal) posterior probability $q(\mathbf{x}_{t-1} \mid \mathbf{x}_t)$. Following this, we prove that *diffusion models have arbitrarily large denoising errors for approximating some common data distributions $q(\mathbf{x}_0)$ (e.g., Gaussian mixture)*, which indicates that some assumption of prior works—bounded score estimation errors—is too strong, which undermines their theoretical guarantees. Lastly and importantly, we prove that *diffusion models will have an arbitrarily large error in matching the learnable backward process $p_\theta(\mathbf{x}_{0:T})$ with the predefined forward process $q(\mathbf{x}_{0:T})$*, even though matching these is the very optimization objective of current diffusion models (Ho et al., 2020; Song et al., 2021b). This finding indicates that diffusion models might fail to fit complex data distributions.

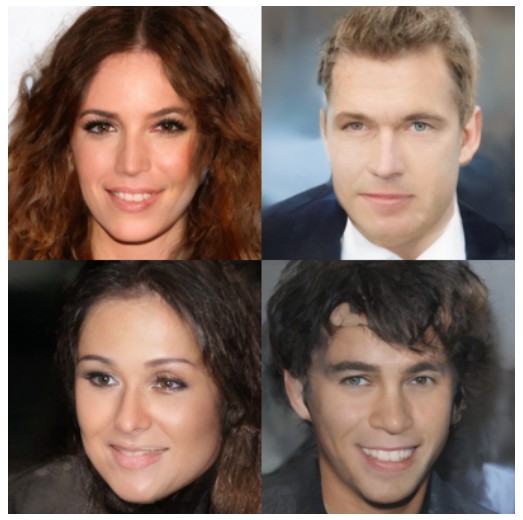 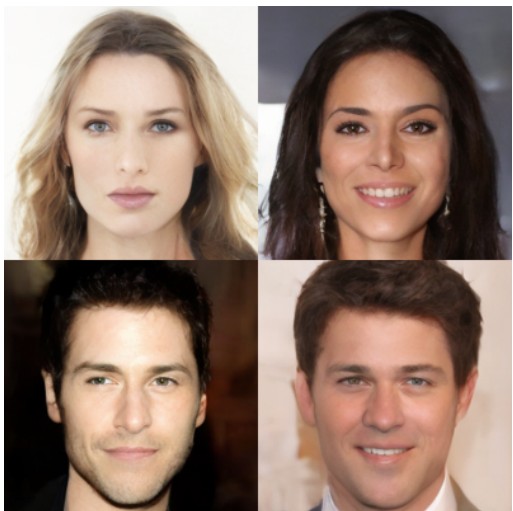

(a) Baseline: vanilla LDM; FID: 11.29.                  (b) Our model: LDM w/ SMD; FID: 6.85.

Figure 1: **SMD improves quality and reduces the number of backward iterations.** Results for CelebA-HQ $256 \times 256$ with *only* 100 *backward iterations*, for LDM with and without SDM. SDM achieves better realism and FID. Achieving the same FID with vanilla LDM would require $8\times$ more steps (see Fig. 3). Note that SMD differs from fast samplers (e.g., DDIM (Song et al., 2021a) and DPM (Lu et al., 2022)): *while those methods focus on deterministic sampling and numerical stability, SMD improves the expressiveness of diffusion models.*

**Our method: Soft Mixture Denoising (SMD).** In light of our theoretical findings, we propose Soft Mixture Denoising (SMD), which aims to represent the hidden mixture components of the posterior probability with a continuous relaxation. We prove that *SMD permits diffusion models to accurately approximate any Gaussian mixture distributions*. For efficiency, we reparameterize SMD and derive an upper bound of the negative log-likelihood for optimization. All in all, this provides a new backward denoising paradigm to the diffusion models that improves expressiveness and permits few backward iterations, yet retains tractability.

**Contributions.** In summary, our contributions are threefold:

1. In terms of theory, we find that current diffusion models suffer from an *expressive bottleneck*. We prove that the models have unbounded errors in both local and global denoising, demonstrating that the assumption of *bounded score estimation errors* made by current theoretical guarantees is too strong;

2. In terms of methodology, we introduce SMD, an expressive backward denoising model. Not only does SMD permit the diffusion models to accurately fit Gaussian mixture distributions, but it is also simple and efficient to implement;

3. In terms of experiments, we show that SMD significantly improves the generation quality of different diffusion models (DDPM (Ho et al., 2020), DDIM (Song et al., 2021a), ADM (Dhariwal & Nichol, 2021), and LDM (Rombach et al., 2022)), especially for few backward iterations—see Fig. 1 for a preview. Since SMD lets diffusion models achieve competitive performances at a smaller number of denoising steps, it can speed up sampling and reduce the cost of existing models.

## 2 BACKGROUND: DISCRETE-TIME DIFFUSION MODELS

In this section, we briefly review the mainstream architecture of diffusion models in discrete time (e.g., DDPM (Ho et al., 2020)). The notations and terminologies introduced below are necessary preparations for diving into subsequent sections.

A diffusion model typically consists of two Markov chains of $T$ steps. One of them is the forward process—also known as the diffusion process—which incrementally adds Gaussian noises to the real sample $\mathbf{x}_0 \in \mathbb{R}^D, D \in \mathbb{N}$, giving a chain of variables $\mathbf{x}_{1:T} = [\mathbf{x}_1, \mathbf{x}_2, \cdots, \mathbf{x}_T]$:

$$q(\mathbf{x}_{1:T} \mid \mathbf{x}_0) = \prod_{t=1}^{T} q(\mathbf{x}_t \mid \mathbf{x}_{t-1}), \quad q(\mathbf{x}_t \mid \mathbf{x}_{t-1}) = \mathcal{N}(\mathbf{x}_t; \sqrt{1 - \beta_t}\mathbf{x}_{t-1}, \beta_t \mathbf{I}), \tag{1}$$

where $\mathcal{N}$ denotes a Gaussian distribution, $\mathbf{I}$ represents an identity matrix, and $\beta_t, 1 \leqslant t \leqslant T$ are a predefined variance schedule. By properly defining the variance schedule, the last variable $\mathbf{x}_T$ will approximately follow a normal Gaussian distribution.

The second part of diffusion models is the *backward* (or *reverse*) *process*. Specifically speaking, the process first draws an initial sample $\mathbf{x}_T$ from a standard Gaussian $p(\mathbf{x}_T) = \mathcal{N}(\mathbf{0}, \mathbf{I})$ and then gradually denoises it into a sequence of variables $\mathbf{x}_{T-1:0} = [\mathbf{x}_{T-1}, \mathbf{x}_{T-2}, \cdots, \mathbf{x}_0]$:

$$p_\theta(\mathbf{x}_{T:0}) = p(\mathbf{x}_T) \prod_{t=T}^{1} p_\theta(\mathbf{x}_{t-1} \mid \mathbf{x}_t), \quad p_\theta(\mathbf{x}_{t-1} \mid \mathbf{x}_t) = \mathcal{N}(\mathbf{x}_{t-1}; \boldsymbol{\mu}_\theta(\mathbf{x}_t, t), \sigma_t \mathbf{I}), \tag{2}$$

where $\sigma_t \mathbf{I}$ is a predefined covariance matrix and $\boldsymbol{\mu}_\theta$ is a learnable module with the parameter $\theta$ to predict the mean vector. Ideally, the learnable backward probability $p_\theta(\mathbf{x}_{t-1} \mid \mathbf{x}_t)$ is equal to the inverse forward probability $q(\mathbf{x}_{t-1} \mid \mathbf{x}_t)$ at every iteration $t \in [1, T]$ such that the backward process is just a reverse version of the forward process.

Since the exact negative log-likelihood $\mathbb{E}[-\log p_\theta(\mathbf{x}_0)]$ is computationally intractable, common practices adopt its upper bound $\mathcal{L}$ as the loss function

$$\mathbb{E}_{\mathbf{x}_0 \sim q(\mathbf{x}_0)}[-\log p_\theta(\mathbf{x}_0)] \leqslant \underbrace{\mathbb{E}_q[\mathcal{D}_{\text{KL}}[q(\mathbf{x}_T \mid \mathbf{x}_0), p(\mathbf{x}_T)]]}_{\mathcal{L}_T} + \underbrace{\mathbb{E}_q[-\log p_\theta(\mathbf{x}_0 \mid \mathbf{x}_1)]}_{\mathcal{L}_0}$$
$$+ \sum_{1 < t \leqslant T} \underbrace{\mathbb{E}_q[\mathcal{D}_{\text{KL}}[q(\mathbf{x}_{t-1} \mid \mathbf{x}_t, \mathbf{x}_0), p_\theta(\mathbf{x}_{t-1} \mid \mathbf{x}_t)]]}_{\mathcal{L}_{t-1}} = \mathcal{L}, \tag{3}$$

where $\mathcal{D}_{\text{KL}}$ denotes the KL divergence. Every term of this loss has an analytic form so that it is computationally optimizable. Ho et al. (2020) further applied some reparameterization tricks to the loss $\mathcal{L}$ for reducing its variance. As a result, the module $\boldsymbol{\mu}_\theta$ is reparameterized as

$$\boldsymbol{\mu}_\theta(\mathbf{x}_t, t) = \frac{1}{\sqrt{\alpha_t}}\left(\mathbf{x}_t - \frac{\beta_t}{\sqrt{1 - \bar{\alpha}_t}}\boldsymbol{\epsilon}_\theta(\mathbf{x}_t, t)\right), \tag{4}$$

where $\alpha_t = 1 - \beta_t$, $\bar{\alpha}_t = \prod_{t'=1}^{t} \alpha_{t'}$, and $\boldsymbol{\epsilon}_\theta$ is parameterized by neural networks. Under this popular scheme, the loss $\mathcal{L}$ is finally simplified as

$$\mathcal{L} = \sum_{t=1}^{T} \mathbb{E}_{\mathbf{x}_0 \sim q(\mathbf{x}_0), \boldsymbol{\epsilon} \sim \mathcal{N}(\mathbf{0}, \mathbf{I})}\left[\|\boldsymbol{\epsilon} - \boldsymbol{\epsilon}_\theta(\sqrt{\bar{\alpha}_t}\mathbf{x}_0 + \sqrt{1 - \bar{\alpha}_t}\boldsymbol{\epsilon}, t)\|^2\right], \tag{5}$$

where the denoising function $\boldsymbol{\epsilon}_\theta$ is tasked to fit Gaussian nosie $\boldsymbol{\epsilon}$.

## 3 THEORY: DMS SUFFER FROM AN EXPRESSIVE BOTTLENECK

In this section, we first show that the Gaussian denoising paradigm leads to an *expressive bottleneck* for diffusion models to fit multimodal data distribution $q(\mathbf{x}_0)$. Then, we properly define two errors $\mathcal{M}_t, \mathcal{E}$ that measure the approximation capability of general diffusion models and prove that they can both be unbounded for current models.

### 3.1 LIMITED GAUSSIAN DENOISING

The core of diffusion models is to let the learnable backward probability $p_\theta(\mathbf{x}_{t-1} \mid \mathbf{x}_t)$ at every iteration $t$ fit the posterior forward probability $q(\mathbf{x}_{t-1} \mid \mathbf{x}_t)$. From Eq. (2), we see that the learnable probability is configured as a simple Gaussian $\mathcal{N}(\mathbf{x}_{t-1}; \boldsymbol{\mu}_\theta(\mathbf{x}_t, t), \sigma_t \mathbf{I})$. While this setup is analytically tractable and computationally efficient, our proposition below shows that its approximation goal $q(\mathbf{x}_{t-1} \mid \mathbf{x}_t)$ might be much more complex.

**Proposition 3.1** (Non-Gaussian Inverse Probability). *For the diffusion process defined in Eq. (1), suppose that the real data follow a Gaussian mixture:* $q(\mathbf{x}_0) = \sum_{k=1}^{K} w_k \mathcal{N}(\mathbf{x}_0; \boldsymbol{\mu}_k, \boldsymbol{\Sigma}_k)$, *which consists of $K$ Gaussian components with mixture weight $w_k$, mean vector $\boldsymbol{\mu}_k$, and covariance matrix $\boldsymbol{\Sigma}_k$, then the posterior forward probability $q(\mathbf{x}_{t-1} \mid \mathbf{x}_t)$ at every iteration $t \in [1, T]$ is another mixture of Gaussian distributions:*

$$q(\mathbf{x}_{t-1} \mid \mathbf{x}_t) = \sum_{k=1}^{K} w'_k \mathcal{N}(\mathbf{x}_{t-1}; \boldsymbol{\mu}'_k, \boldsymbol{\Sigma}'_k), \tag{6}$$

*where $w'_k, \boldsymbol{\mu}'_k$ depend on both variable $\mathbf{x}_t$ and $\boldsymbol{\mu}_t$.*

*Remark* 3.1. The Gaussian mixture in theory is a universal approximator of smooth probability densities (Dalal & Hall, 1983; Goodfellow et al., 2016). Therefore, this proposition implies that the posterior forward probability $q(\mathbf{x}_{t-1} \mid \mathbf{x}_t)$ can be arbitrarily complex.

*Proof.* The proof to this proposition is fully provided in Appendix B. $\square$

While diffusion models perform well in practice, we can infer from above that the Gaussian denoising paradigm $p_\theta(\mathbf{x}_{t-1} \mid \mathbf{x}_t) = \mathcal{N}(\mathbf{x}_{t-1}; \boldsymbol{\mu}_\theta(\mathbf{x}_t, t), \sigma_t \mathbf{I})$ causes a bottleneck for the backward probability to fit the potentially multimodal distribution $q(\mathbf{x}_{t-1} \mid \mathbf{x}_t)$. Importantly, this problem is not rare since real-world data distributions are commonly non-Gaussian and multimodal. For example, classes in a typical image dataset are likely to form separate modes, and possibly even multiple modes per class (e.g. different dog breeds).

> **Takeaway**: The posterior forward probability $q(\mathbf{x}_{t-1} \mid \mathbf{x}_t)$ can be arbitrarily complex for the Gaussian backward probability $p_\theta(\mathbf{x}_{t-1} \mid \mathbf{x}_t) = \mathcal{N}(\mathbf{x}_{t-1}; \boldsymbol{\mu}_\theta(\mathbf{x}_t, t), \sigma_t \mathbf{I})$ to approximate. We call this problem the *expressive bottleneck* of diffusion models.

## 3.2 Denoising and Approximation Errors

To quantify the impact of this expressive bottleneck, we define two error measures in terms of local and global denoising errors, i.e., the discrepancy between forward process $q(\mathbf{x}_{0:T})$ and backward process $p_\theta(\mathbf{x}_{0:T})$.

**Derivation of the local denoising error.** Considering the form of loss term $\mathcal{L}_{t-1}$ in Eq. (3), we apply the KL divergence to estimate the approximation error of every learnable backward probability $p_\theta(\mathbf{x}_{t-1} \mid \mathbf{x}_t), t \in [1, T]$ to its reference $q(\mathbf{x}_{t-1} \mid \mathbf{x}_t)$ as $\mathcal{D}_{\mathrm{KL}}[q(\mathbf{x}_{t-1} \mid \mathbf{x}_t), p_\theta(\mathbf{x}_{t-1} \mid \mathbf{x}_t)]$. In Appendix A, we prove that the single-step backward model $p_\theta(\mathbf{x}_{t-1} \mid \mathbf{x}_t)$ is optimized towards $q(\mathbf{x}_{t-1} \mid \mathbf{x}_t)$ for training with loss $\mathcal{L}$. Since the error depends on variable $\mathbf{x}_t$, we normalize it with density $q(\mathbf{x}_t)$ into $\mathbb{E}[\mathcal{D}_{\mathrm{KL}}[\cdot]] = \int_{\mathbf{x}_t} q(\mathbf{x}_t) \mathcal{D}_{\mathrm{KL}}[\cdot] d\mathbf{x}_t$. Importantly, we take the infimum of this error over the parameter space $\Theta$ as $\inf_{\theta \in \Theta}(\int_{\mathbf{x}_t} q(\mathbf{x}_t) \mathcal{D}_{\mathrm{KL}}[q(\cdot), p_\theta(\cdot)] d\mathbf{x}_t)$, which means neural networks are globally optimized. In light of the above derivation, we have the below definition.

**Definition 3.1** (Local Denoising Error). For every learnable backward probability $p_\theta(\mathbf{x}_{t-1} \mid \mathbf{x}_t), 1 \leq t \leq T$ in a diffusion model, its error of best approximation (i.e., parameter $\theta$ is globally optimized) to the reference $q(\mathbf{x}_{t-1} \mid \mathbf{x}_t)$ is defined as

$$\mathcal{M}_t = \inf_{\theta \in \Theta} \left( \mathbb{E}_{\mathbf{x}_t \sim q(\mathbf{x}_t)}[\mathcal{D}_{\mathrm{KL}}[q(\mathbf{x}_{t-1} \mid \mathbf{x}_t), p_\theta(\mathbf{x}_{t-1} \mid \mathbf{x}_t)]] \right)$$
$$= \inf_{\theta \in \Theta} \left( \int_{\mathbf{x}_t} \underbrace{q(\mathbf{x}_t)}_{\text{Density Weight}} \underbrace{\mathcal{D}_{\mathrm{KL}}[q(\mathbf{x}_{t-1} \mid \mathbf{x}_t), p_\theta(\mathbf{x}_{t-1} \mid \mathbf{x}_t)]}_{\text{Denoising Error w.r.t. the Input } \mathbf{x}_t} d\mathbf{x}_t \right), \tag{7}$$

where space $\Theta$ represents the set of all possible parameters. Note that the inequality $\mathcal{M}_t \geq 0$ always holds because KL divergence is non-negative.

**Significance of the global denoising error.** Current practices (Ho et al., 2020) expect the backward process $p_\theta(\mathbf{x}_{0:T})$ to exactly match the forward process $q(\mathbf{x}_{0:T})$ such that their marginals at iteration 0 are equal: $q(\mathbf{x}_0) = p_\theta(\mathbf{x}_0)$. For example, Song et al. (2021b) directly configured the backward process as the reverse-time diffusion equation. Hence, we have the following error definition to measure the global denoising capability of diffusion models.

**Definition 3.2** (Global Denoising Error). The discrepancy between learnable backward process $p_\theta(\mathbf{x}_{0:T})$ and predefined forward process $q(\mathbf{x}_{0:T})$ is estimated as

$$\mathcal{E} = \inf_{\theta \in \Theta} \Big( \mathcal{D}_{\mathrm{KL}}[q(\mathbf{x}_{0:T}), p_\theta(\mathbf{x}_{0:T})] \Big), \tag{8}$$

where again $\mathcal{E} \geqslant 0$ always holds since KL divergence is non-negative.

### 3.3 LIMITED APPROXIMATION THEOREMS

In this part, we prove that the above defined errors are unbounded for current diffusion models.[1]

**Theorem 3.1** (Uniformly Unbounded Denoising Error). *For the diffusion process defined in Eq. (1) and the Gaussian denoising process defined in Eq. (2), there exists a continuous data distribution $q(\mathbf{x}_0)$ (more specifically, Gaussian mixture) such that $\mathcal{M}_t$ is uniformly unbounded—given any real number $N \in \mathbb{R}$, the inequality $\mathcal{M}_t > N$ holds for every denoising iteration $t \in [1, T]$.*

*Proof.* We provide a complete proof to this theorem in Appendix C. □

The above theorem not only implies that current diffusion models fail to fit some multimodal data distribution $q(\mathbf{x}_t)$ because of their limited expressiveness in local denoising, but also indicates that the assumption of *bounded score estimation errors* (i.e., bounded denoising errors) is too strong. Consequently, this undermines existing theoretical guarantees (Lee et al., 2022a; Chen et al., 2023) that aim to prove that diffusion models are universal approximates.

> *Takeaway*: The denoising error $\mathcal{M}_t$ of current diffusion models can be arbitrarily large at every denoising step $t \in [1, T]$. Thus, the assumption of *bounded score estimation errors* made by existing theoretical guarantees is too strong.

Based on Theorem 3.1 and Proposition 3.1, we finally show that the global denoising error $\mathcal{E}$ of current diffusion models is also unbounded.

**Theorem 3.2** (Unbounded Approximation Error). *For the forward and backward processes respectively defined in Eq. (1) and Eq. (2), given any real number $N \in \mathbb{R}$, there exists a continuous data distribution $q(\mathbf{x}_0)$ (specifically, Gaussian mixture) such that $\mathcal{E} > N$.*

*Proof.* A complete proof to this theorem is offered in Appendix D. □

Since the negative likelihood $\mathbb{E}[-\log p_\theta(\mathbf{x}_0)]$ is computationally feasible, current practices (e.g., DDPM (Ho et al., 2020) and SGM (Song et al., 2021b)) optimize the diffusion models by matching the backward process $p_\theta(\mathbf{x}_{0:T})$ with the forward process $q(\mathbf{x}_{0:T})$. This theorem indicates that this optimization scheme will fail for some complex data distribution $q(\mathbf{x}_0)$.

**Why diffusion models already perform well in practice.** The above theorem may bring unease—how can this be true when diffusion models are considered highly-realistic data generators? The key lies in the number of denoising steps. As indicated in Eq. (29), the more steps are used, the more the backward probability, Eq. (2), is centered around a single mode, hence the more the simple Gaussian assumption holds (Sohl-Dickstein et al., 2015). As a result, we will see in Sec. 5.3 that our own method, which makes no Gaussian posterior assumption, improves quality especially for few backward iterations.

> *Takeaway*: Standard diffusion models (e.g. DDPM) with simple Gaussian denoising poorly approximate some multimodal distributions (e.g. Gaussian mixture). This is problematic, as these distributions are very common in practice.

---

[1] It is also worth noting that these errors already overestimate the performances of diffusion models, since their definitions involve an infimum operation $\inf_{\theta \in \Theta}$.

## 4 METHOD: SOFT MIXTURE DENOISING

Our theoretical studies showed how current diffusion models have limited expressiveness to approximate multimodal data distributions. To solve this problem, we propose **soft mixture denoising** (SMD), a tractable relaxation of a Gaussian mixture model for modelling the denoising posterior.

### 4.1 MAIN THEORY

Our theoretical analysis highlight an expressive bottleneck of current diffusion models due to its Gaussian denoising assumption. Based on Proposition 3.1, an obvious way to address this problem is to directly model the backward probability $p_\theta(\mathbf{x}_{t-1} \mid \mathbf{x}_t)$ as a Gaussian mixture. For example, we could model:

$$p_\theta^{\mathrm{mixture}}(\mathbf{x}_{t-1} \mid \mathbf{x}_t) = \sum_{k=1}^{K} z_{\theta_k}(\mathbf{x}_t, t) \mathcal{N}(\mathbf{x}_{t-1}; \boldsymbol{\mu}_{\theta_k}(\mathbf{x}_t, t), \boldsymbol{\Sigma}_{\theta_k}(\mathbf{x}_t, t)), \qquad (9)$$

where $\theta = \bigcup_{k=1}^{K} \theta_k$, the number of Gaussian components $K$ is a hyperparameter, and where weight $z_t^k(\cdot)$, mean $\boldsymbol{\mu}_{\theta_k}^k(\cdot)$, and covariance $\boldsymbol{\Sigma}_{\theta_k}^k(\cdot)$ are learnable and determine each of the mixture components. While the mixture model might be complex enough for backward denoising, it is not practical for two reasons: 1) it is often intractable to determine the number of components $K$ from observed data; 2) mixture models are notoriously hard to optimize. Actually, Jin et al. (2016) proved that a Gaussian mixture model might be optimized into an arbitrarily bad local optimum.

**Soft mixture denoising.** To efficiently improve the expressiveness of diffusion models, we introduce *soft mixture denoising* (SMD) $p_{\bar{\theta}}^{\mathrm{SMD}}(\mathbf{x}_{t-1} \mid \mathbf{x}_t)$, a soft version of the mixture model $p_\theta^{\mathrm{mixture}}(\cdot)$, which avoids specifying the number of mixture components $K$ and permits effective optimization. Specifically, we define a continuous latent variable $\mathbf{z}_t$, as an alternative to mixture weight $z_t^k$, that represents the potential mixture structure of posterior distribution $q(\mathbf{x}_{t-1} \mid \mathbf{x}_t)$. Under this scheme, we model the learnable backward probability as

$$p_{\bar{\theta}}^{\mathrm{SMD}}(\cdot) = \int p_{\bar{\theta}}^{\mathrm{SMD}}(\mathbf{x}_{t-1}, \mathbf{z}_t \mid \mathbf{x}_t) d\mathbf{z}_t = \int p_{\bar{\theta}}^{\mathrm{SMD}}(\mathbf{z}_t \mid \mathbf{x}_t) p_{\bar{\theta}}^{\mathrm{SMD}}(\mathbf{x}_{t-1} \mid \mathbf{x}_t, \mathbf{z}_t) d\mathbf{z}_t, \qquad (10)$$

where $\bar{\theta}$ denotes the set of all learnable parameters. We model $p_{\bar{\theta}}(\mathbf{x}_{t-1} \mid \mathbf{x}_t, \mathbf{z}_t)$ as a learnable multivariate Gaussian and expect that different values of the latent variable $\mathbf{z}_t$ will correspond to differently parameterized Gaussians:

$$p_{\bar{\theta}}^{\mathrm{SMD}}(\mathbf{x}_{t-1} \mid \mathbf{x}_t, \mathbf{z}_t) = \mathcal{N}\big(\mathbf{x}_{t-1}; \boldsymbol{\mu}_{\theta \bigcup f_\phi(\mathbf{z}_t, t)}(\mathbf{x}_t, t), \boldsymbol{\Sigma}_{\theta \bigcup f_\phi(\mathbf{z}_t, t)}(\mathbf{x}_t, t)\big), \qquad (11)$$

where $\theta \subset \bar{\theta}$ is a set of vanilla learnable parameters and $f_\phi(\mathbf{z}_t, t)$ is another collection of parameters computed from a neural network $f_\phi$ with learnable parameters $\phi \subset \bar{\theta}$. Both $\theta$ and $f_\phi(\mathbf{z}_t, t)$ constitute the parameter set of mean and covariance functions $\boldsymbol{\mu}_\bullet, \boldsymbol{\Sigma}_\bullet$ for computations, but only $\theta$ and $\phi$ will be optimized. This type of design is similar to the hypernetwork (Ha et al., 2017; Krueger et al., 2018). For implementation, we follow Eq. (2) to constrain the covariance matrix $\boldsymbol{\Sigma}_\bullet$ to the form $\sigma_t \mathbf{I}$ and parameterize mean $\boldsymbol{\mu}_\bullet(\mathbf{x}_t, t)$ similar to Eq. (4):

$$\boldsymbol{\mu}_{\theta \bigcup f_\phi(\mathbf{z}_t, t)}(\mathbf{x}_t, t) = \frac{1}{\sqrt{\alpha_t}}\Big(\mathbf{x}_t - \frac{\beta_t}{\sqrt{1 - \bar{\alpha}_t}} \boldsymbol{\epsilon}_{\theta \bigcup f_\phi(\mathbf{z}_t, t)}(\mathbf{x}_t, t)\Big), \qquad (12)$$

where $\boldsymbol{\epsilon}_\bullet$ is a neural network. For image data, we build it as a U-Net (Ronneberger et al., 2015) (i.e., $\theta$) with several extra layers that are computed from $f_\phi(\mathbf{z}_t, t)$.

For the mixture component $p_{\bar{\theta}}(\mathbf{z}_t \mid \mathbf{x}_t)$, we parameterize it with a neural network such that it can be an arbitrarily complex distribution and adds great flexibility into the backward probability $p_{\bar{\theta}}^{\mathrm{SMD}}(\mathbf{x}_{t-1} \mid \mathbf{x}_t)$. For implementation, we adopt a mapping $g_\xi : (\boldsymbol{\eta}, \mathbf{x}_t, t) \mapsto \mathbf{z}_t, \xi \subset \bar{\theta}$ with $\boldsymbol{\eta} \overset{\mathrm{i.i.d.}}{\sim} \mathcal{N}(\mathbf{0}, \mathbf{I})$, which converts a standard Gaussian into a non-Gaussian distribution.

**Theoretical guarantee.** We prove that SMD $p_{\bar{\theta}}^{\mathrm{SMD}}(\mathbf{x}_{t-1} \mid \mathbf{x}_t)$ improves the expressiveness of diffusion models—resolving the limitations highlighted in Theorems 3.1 and 3.2.

**Theorem 4.1** (Expressive Soft Mixture Denoising). *For the diffusion process defined in Eq. (1), suppose soft mixture model $p_{\bar{\theta}}^{\mathrm{SMD}}(\mathbf{x}_{t-1} \mid \mathbf{x}_t)$ is applied for backward denoising and data distribution $q(\mathbf{x}_0)$ is a Gaussian mixture, then both $\mathcal{M}_t = 0, \forall t \in [1, T]$ and $\mathcal{E} = 0$ hold.*

| **Algorithm 1** Training | **Algorithm 2** Sampling |
|---|---|
| 1: **repeat** | 1: $\mathbf{x}_T \sim p(\mathbf{x}_T) = \mathcal{N}(\mathbf{0}, \mathbf{I})$ |
| 2:  $\mathbf{x}_0 \sim q(\mathbf{x}_0)$ | 2: **for** $t = T, \ldots, 1$ **do** |
| 3:  $t \sim \mathcal{U}\{1, T\},\ \boldsymbol{\epsilon} \sim \mathcal{N}(\mathbf{0}, \mathbf{I})$ | 3:  $\boldsymbol{\epsilon} \sim \mathcal{N}(\mathbf{0}, \mathbf{I})$ if $t > 1$, else $\boldsymbol{\epsilon} = 0$ |
| 4:  $\mathbf{x}_t = \sqrt{\bar{\alpha}_t}\mathbf{x}_0 + \sqrt{1 - \bar{\alpha}_t}\boldsymbol{\epsilon}$ | 4:  $\boldsymbol{\eta} \sim \mathcal{N}(\mathbf{0}, \mathbf{I})$ |
| 5:  $\boldsymbol{\eta} \sim \mathcal{N}(\mathbf{0}, \mathbf{I})$ | 5:  Latent variable sampling: $\mathbf{z}_t = g_\xi(\boldsymbol{\eta}, \mathbf{x}_t, t)$ |
| 6:  Latent variable sampling: $\mathbf{z}_t = g_\xi(\boldsymbol{\eta}, \mathbf{x}_t, t)$ | 6:  Param. for computation: $\hat{\theta} = \theta \bigcup f_\phi(\mathbf{z}_t, t)$ |
| 7:  Param. for computation: $\hat{\theta} = \theta \bigcup f_\phi(\mathbf{z}_t, t)$ | 7:  $\mathbf{x}_{t-1} = \frac{1}{\sqrt{\alpha_t}}\left(\mathbf{x}_t - \frac{1-\alpha_t}{\sqrt{1-\bar{\alpha}_t}}\ \boldsymbol{\epsilon}_{\hat{\theta}}\left(\mathbf{x}_t, t\right)\right) + \sigma_t \boldsymbol{\epsilon}$ |
| 8:  Param. to optimize: $\bar{\theta} = \theta \bigcup \phi \bigcup \xi$ | 8: **end for** |
| 9:  Update $\bar{\theta}$ w.r.t. $\nabla_{\bar{\theta}} \|\boldsymbol{\epsilon} - \boldsymbol{\epsilon}_{\hat{\theta}}(\mathbf{x}_t, t)\|^2$ | 9: **return** $\mathbf{x}_0$ |
| 10: **until** converged | |

*Proof.* The proof to this theorem is fully provided in Appendix E. □

*Remark* 4.1. The Gaussian mixture is a universal approximator for continuous probability distributions (Dalal & Hall, 1983). Therefore, this theorem implies that our proposed SMD permits the diffusion models to well approximate arbitrarily complex data distributions.

> ***Takeaway***: Soft mixture denoising (SMD) parameterizes the backward probability as a continuously relaxed Gaussian mixture, which potentially permits the diffusion models to well approximate any continuous data distribution.

### 4.2 EFFICIENT OPTIMIZATION AND SAMPLING

While Theorem 4.1 shows that SMDs are highly expressive, it assumes the neural networks are globally optimized. Plus, the latent variable in SMD introduces more complexity to the computation and analysis of diffusion models. To fully exploit the potential of SMD, we thus need efficient optimization and sampling algorithms.

**Loss function.** The negative log-likelihood for a diffusion model with the backward probability $p_{\bar{\theta}}^{\text{SMD}}(\mathbf{x}_{t-1} \mid \mathbf{x}_t)$ of a latent variable model is formally defined as

$$\mathbb{E}_q[-\ln p_{\bar{\theta}}^{\text{SMD}}(\mathbf{x}_0)] = \mathbb{E}_{\mathbf{x}_0 \sim q(\mathbf{x}_0)}\left[-\ln\left(\int_{\mathbf{x}_{1:T}} p(\mathbf{x}_T) \prod_{t=T}^{1} p_{\bar{\theta}}^{\text{SMD}}(\mathbf{x}_{t-1} \mid \mathbf{x}_t) d\mathbf{x}_{1:T}\right)\right]. \tag{13}$$

Like vanilla diffusion models, this log-likelihood term is also computationally infeasible. In the following, we derive its upper bound for optimization.

**Proposition 4.1** (Upper Bound of Negative Log-likelihood). *Suppose the diffusion process is defined as Eq. (1) and the soft mixture model $p_{\bar{\theta}}^{\text{SMD}}(\mathbf{x}_{t-1} \mid \mathbf{x}_t)$ is applied for backward denoising, then an upper bound of the expected negative log-likelihood $\mathbb{E}_q[-\ln p_{\bar{\theta}}^{\text{SMD}}(\mathbf{x}_0)]$ is*

$$\mathcal{L}^{\text{SMD}} = C + \sum_{t=1}^{T} \mathbb{E}_{\boldsymbol{\eta}, \boldsymbol{\epsilon}, \mathbf{x}_0}\left[\Gamma_t \|\boldsymbol{\epsilon} - \boldsymbol{\epsilon}_{\theta \bigcup f_\phi(g_\xi(\cdot), t)}\left(\sqrt{\bar{\alpha}_t}\mathbf{x}_0 + \sqrt{1-\bar{\alpha}_t}\boldsymbol{\epsilon}, t\right)\|^2\right], \tag{14}$$

*where $g_\xi(\cdot) = g_\xi(\boldsymbol{\eta}, \sqrt{\bar{\alpha}_t}\mathbf{x}_0 + \sqrt{1-\bar{\alpha}_t}\boldsymbol{\epsilon}, t)$, $C$ is a constant that does not involve any learnable parameter $\bar{\theta} = \theta \bigcup \phi \bigcup \xi$, $\mathbf{x}_0 \sim q(\mathbf{x}_0)$, $\boldsymbol{\eta}, \boldsymbol{\epsilon}$ are two independent variables drawn from standard Gaussians, and $\Gamma_t = \beta_t^2/(2\sigma_t\alpha_t(1-\bar{\alpha}_t))$.*

*Proof.* The detailed derivation to get the upper bound $\mathcal{L}^{\text{SMD}}$ is in Appendix F. □

Compared with the loss function of vanilla diffusion models, Eq. (5), our upper bound mainly differs in the hypernetwork $f_\phi$ to parameterize the denoising function $\boldsymbol{\epsilon}_\bullet$ and an expectation operation $\mathbb{E}_{\boldsymbol{\eta}}$.

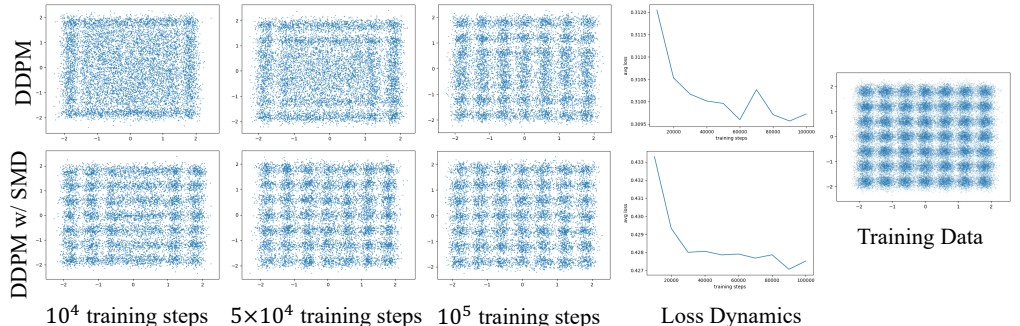

Figure 2: **Visualising the expressive bottleneck of standard diffusion models.** Experimental results on synthetic dataset with $7 \times 7$ Gaussians (right), for DDPM with $T = 1000$. Even though DDPM has converged, we observe that the modes are not easily distinguishable. On the other hand, SMD converges much faster and results in distinguishable modes.

Table 1: **SMD consistently improves generation quality.** FID score of different models across common image datasets and resolutions. We use $T = 1000$ for all models.

| Dataset / Model | DDPM | DDPM w/ SMD | ADM | ADM w/ SMD |
|---|---|---|---|---|
| CIFAR-10 ($32 \times 32$) | 3.78 | **3.13** | 2.98 | **2.55** |
| LSUN-Conference ($64 \times 64$) | 4.15 | **3.52** | 3.85 | **3.29** |
| LSUN-Church ($64 \times 64$) | 3.65 | **3.17** | 3.41 | **2.98** |
| CelebA-HQ ($128 \times 128$) | 6.78 | **6.35** | 6.45 | **6.02** |

The former is computed by neural networks and the latter is approximated by Monte Carlo sampling, which both add minor computational costs.

**Training and Inference.** The SMD training and sampling procedures are respectively shown in Algorithms 1 and 2, with blue highlighting differences with vanilla diffusion. For the training procedure, we follow common practices of (Ho et al., 2020; Dhariwal & Nichol, 2021), and (1) apply Monte Carlo sampling to handle iterated expectations $\mathbb{E}_{\boldsymbol{\eta},\boldsymbol{\epsilon},\mathbf{x}_0}$ in Eq. (14), and (2) reweigh loss term $\|\boldsymbol{\epsilon} - \boldsymbol{\epsilon}_\bullet(\mathbf{x}_t, t)\|^2$ by ignoring coefficient $\Gamma_t$. One can also sample more noises (e.g., $\boldsymbol{\eta}$) in one training step to trade run-time efficiency for approximation accuracy. The source code of this work is publicly available at a personal repository: https://github.com/louisli321/smd, and our lab repository: https://github.com/vanderschaarlab/smd.

## 5 EXPERIMENTS

Let us verify how SMD improves the quality and speed of existing diffusion models. First, we use a toy example to visualise that existing diffusion models struggle to learn multivariate Gaussians, whereas SMD does not. Subsequently, we show how SMD significantly improves the FID score across different types of diffusion models (e.g., DDPM, ADM (Dhariwal & Nichol, 2021), LDM) and datasets. Then, we demonstrate how SMD significantly improves performance at low number of inference steps. This enables reducing the number of inference steps, thereby speeding up generation and reducing computational costs. Lastly, we show how quality can be improved even further by sampling more than one $\boldsymbol{\eta}$ for loss estimation at training time, which further improves the performance but causes an extra time cost.

### 5.1 VISUALISING THE EXPRESSIVE BOTTLENECK

From Proposition 3.1 and Theorems 3.2, 3.1 it follows that vanilla diffusion models would struggle with learning a Gaussian Mixture model, whereas Theorem 4.1 proves SMD does not. Let us visualise this difference using a simple toy experiment. In Figure 2 we plot the learnt distribution of DDPM over the training process, with and without SMD. We observe that DDPM with SMD converges much faster, and provides a more accurate distribution at time of convergence.

## 5.2 SMD IMPROVES IMAGE QUALITY

We select three of the most common diffusion models and four image datasets to show how our proposed SMD quantitatively improves diffusion models. Baselines include DDPM Ho et al. (2020), ADM (Dhariwal & Nichol, 2021), and Latent Diffusion Model (LDM) (Pinaya et al., 2022). Datasets include CIFAR-10 (Krizhevsky et al., 2009), LSUN-Conference, LSUN-Church (Yu et al., 2015), and CelebA-HQ (Liu et al., 2015). For all models, we set the backward iterations $T$ as 1000 and generate 10000 images for computing FID scores.

In Table 1, we show how the proposed SMD significantly improves both DDPM and ADM on all datasets, for a range of resolutions. For example, SDM outperforms DDPM by $15.14\%$ on LSUN-Church and ADM by $16.86\%$. Second, in Table 2 we include results for high-resolution image datasets, see

Table 2: **SMD improves LDM generation quality.** FID score of latent diffusion with and without SMD on high-resolution image datasets ($T = 1000$).

| Dataset / Model | LDM | LDM w/ SMD |
|---|---|---|
| LSUN-Church ($256 \times 256$) | 5.86 | **5.21** |
| CelebA-HQ ($256 \times 256$) | 6.13 | **5.48** |

Fig. 1 for example images ($T = 100$). Here we employed LDM as baseline to reduce memory footprint, where we use a pretrained and frozen VAE. We observe that SMD improves FID scores significantly. These results strongly indicate how SMD is effective in improving the performance for different baseline diffusion models.

## 5.3 SMD IMPROVES INFERENCE SPEED

Intuitively, for few denoising iterations the distribution $q(\mathbf{x}_{t-1} \mid \mathbf{x}_t)$ is more of a mixture, which leads to the backward probability $p_\theta(\mathbf{x}_{t-1} \mid \mathbf{x}_t)$—a simple Gaussian—being a worse approximation. Based on Theorems 3.2 and 4.1, we anticipate that our models will be more robust to this effect than vanilla diffusion models.

The solid blue and red curves in Fig. 3 respectively show how the F1 scores of vanilla LDM and LDM w/ SMD change with respect to increasing backward iterations. We can see that our proposed SMD improves the LDM much more at fewer backward iterations (e.g., $T = 200$). We also include LDM with DDIM (Song et al., 2021a), a popular fast sampler. We see that the advantage of SDM is consistent across samplers.

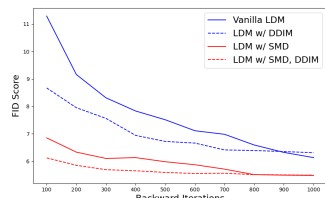

Figure 3: **SMD reduces the number of sampling steps.** Latent DDIM and DDPM for different iterations on CelebA-HQ ($256 \times 256$).

## 5.4 SAMPLING MULTIPLE $\eta$: QUALITY-COST TRADE-OFF

In Algorithm 1, we only sample one $\boldsymbol{\eta}$ at a time for maintaining high computational efficiency. We can sample multiple $\eta$ to estimate the loss better. Figure 4 shows how the training time of one training step and FID score of DDPM with SMD changes as a function of the number of $\eta$ samples. While the time cost linearly goes up with the increasing sampling times, FID monotonically decreases ($6.5\%$ for 5 samples).

## 6 FUTURE WORK

We have proven that there exists an expressive bottleneck in popular diffusion models. Since multimodal distributions are so common, this limitation does matter across domains (e.g., tabular, images, text). Our proposed SMD, as a general method for expressive backward denoising, solves this problem. Regardless of network architectures, SMD can be extended to other tasks, including text-to-image translation, video generation, and speech synthesis. Because SMD provides better quality for fewer steps, we also hope it will become a standard part of diffusion libraries, speeding up both training and inference.

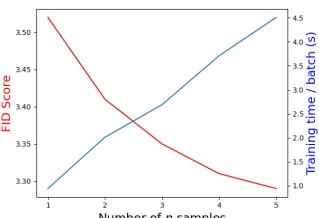

Figure 4: **SMD quality is further improved by sampling multiple $\eta$,** see Alg. 1 on LSUN-Conference ($64 \times 64$) for DDPM w/ SMD.

## ACKNOWLEDGMENTS

We thank the anonymous ICLR reviewers for their kind and constructive reviews. Yangming Li also thanks Accenture for their sponsorship and support.

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

## A    SIGNIFICANCE OF THE LOCAL ERROR $\mathcal{M}_t$

In this part, we show that the backward model $p_\theta(t-1 \mid t), t \in [1,T]$ is optimized towards the inverse probability $q(t-1 \mid t)$ in terms of the loss function $\mathcal{L}$ as defined in Eq. (3).

**Proposition A.1.** *For a perfectly optimized reverse process $\{p_\theta(t-1 \mid t)\}_{t\in[1,T]}$ (as defined in Eq. (2)) that minimizes the loss function $\mathcal{L}$ (as defined in Eq. (3)), equality $p_\theta(t-1 \mid t) = q(t-1 \mid t)$ holds for every denoising iteration $t \in [1,T]$.*

*Proof.* Following DDPM (Ho et al., 2020) and SGM (Song & Ermon, 2019; Song et al., 2020; Karras et al., 2022), the loss function $\mathcal{L}$ can be formulated as

$$\mathcal{L} = \mathbb{E}_{\mathbf{x}_{0:T}\sim q(\mathbf{x}_{0:T})}\Big[ -\ln \frac{p_\theta(\mathbf{x}_{0:T})}{q(\mathbf{x}_{1:T} \mid \mathbf{x}_0)}\Big] = \mathcal{D}_{\mathrm{KL}}[q(\mathbf{x}_{0:T}), p_\theta(\mathbf{x}_{0:T})] - \mathbb{E}_{\mathbf{x}_0\sim q(\mathbf{x}_0)}[\ln q(\mathbf{x}_0)]. \quad (15)$$

Note that the second expectation term $\mathbb{E}[\cdot]$ is a constant and the first KL-divergence term $\mathcal{D}_{\mathrm{KL}}[\cdot]$ reaches its minimum 0 when $p_\theta(\mathbf{x}_{0:T})$ equals $q(\mathbf{x}_{0:T})$. Therefore, for a perfectly optimized diffusion model, we have $p_\theta(\mathbf{x}_{0:T}) = q(\mathbf{x}_{0:T})$. Then, for every iteration $t \in [1,T]$, we get

$$p_\theta(\mathbf{x}_{t-1}, \mathbf{x}_t) = \int p_\theta(\mathbf{x}_{0:T})d\mathbf{x}_{0:t-2}d\mathbf{x}_{t+1:T} = \int q(\mathbf{x}_{0:T})d\mathbf{x}_{0:t-2}d\mathbf{x}_{t+1:T} = q(\mathbf{x}_{t-1}, \mathbf{x}_t). \quad (16)$$

Similarly, we also have the following equality:

$$p_\theta(\mathbf{x}_t) = \int p_\theta(\mathbf{x}_{0:T})d\mathbf{x}_{0:t-1}d\mathbf{x}_{t+1:T} = \int q(\mathbf{x}_{0:T})d\mathbf{x}_{0:t-1}d\mathbf{x}_{t+1:T} = q(\mathbf{x}_t). \quad (17)$$

Based on the above two equations, we finally derive

$$p_\theta(\mathbf{x}_{t-1} \mid \mathbf{x}_t) = \frac{p_\theta(\mathbf{x}_{t-1}, \mathbf{x}_t)}{p_\theta(\mathbf{x}_t)} = \frac{q(\mathbf{x}_{t-1}, \mathbf{x}_t)}{q(\mathbf{x}_t)} = q(\mathbf{x}_{t-1} \mid \mathbf{x}_t), \quad (18)$$

which proves the proposition. $\qquad\square$

The above conclusion indicates that the definition of local denoising error $\mathcal{M}_t$ is proper for its use: quantifying the performance of backward module $p_\theta(t-1 \mid t)$.

## B    PROOF OF PROPOSITION 3.1

By repeatedly applying basic operations (e.g., chain rule) of probability theory to conditional distribution of backward variable $q(\mathbf{x}_{t-1} \mid \mathbf{x}_t)$, we have

$$\begin{aligned} q(\mathbf{x}_{t-1} \mid \mathbf{x}_t) &= \frac{q(\mathbf{x}_t, \mathbf{x}_{t-1})}{q(\mathbf{x}_t)} = \frac{q(\mathbf{x}_t \mid \mathbf{x}_{t-1})q(\mathbf{x}_{t-1})}{q(\mathbf{x}_t)} = \frac{q(\mathbf{x}_t \mid \mathbf{x}_{t-1})}{q(\mathbf{x}_t)}\int_{\mathbf{x}_0} q(\mathbf{x}_{t-1}, \mathbf{x}_0)d\mathbf{x}_0 \\ &= \frac{1}{q(\mathbf{x}_t)}q(\mathbf{x}_t \mid \mathbf{x}_{t-1})\int_{\mathbf{x}_0} q(\mathbf{x}_{t-1} \mid \mathbf{x}_0)q(\mathbf{x}_0)d\mathbf{x}_0 \end{aligned}. \quad (19)$$

Based on Eq. (1) and $q(\mathbf{x}_t \mid \mathbf{x}_0) = \mathcal{N}(\mathbf{x}_t; \sqrt{\bar{\alpha}_t}\mathbf{x}_0, (1-\bar{\alpha}_t)\mathbf{I})$, from (Ho et al., 2020), posterior probability $q(\mathbf{x}_{t-1} \mid \mathbf{x}_t)$ can be expressed as

$$q(\mathbf{x}_{t-1} \mid \mathbf{x}_t) = \frac{\mathcal{N}(\mathbf{x}_t; \sqrt{1-\beta_t}\mathbf{x}_{t-1}, \beta_t\mathbf{I})}{q(\mathbf{x}_t)}\int_{\mathbf{x}_0} \mathcal{N}(\mathbf{x}_{t-1}; \sqrt{\bar{\alpha}_{t-1}}\mathbf{x}_0, (1-\bar{\alpha}_{t-1})\mathbf{I})q(\mathbf{x}_0)d\mathbf{x}_0. \quad (20)$$

Note that for a multivariate Gaussian, the following holds:

$$\begin{aligned} \mathcal{N}(\mathbf{x}; \lambda\boldsymbol{\mu}, \boldsymbol{\Sigma}) &= (2\pi)^{-\frac{D}{2}}|\boldsymbol{\Sigma}|^{-\frac{1}{2}} \exp\Big( -\frac{1}{2}(\mathbf{x} - \lambda\boldsymbol{\mu})^T\boldsymbol{\Sigma}^{-1}(\mathbf{x} - \lambda\boldsymbol{\mu})\Big) \\ &= \frac{1}{\lambda^D}(2\pi)^{-\frac{D}{2}}\Big|\frac{\boldsymbol{\Sigma}}{\lambda^2}\Big|^{-\frac{1}{2}} \exp\Big( -\frac{1}{2}\big(\boldsymbol{\mu} - \frac{\mathbf{x}}{\lambda}\big)^T\big(\frac{\boldsymbol{\Sigma}}{\lambda^2}\big)^{-1}\big(\boldsymbol{\mu} - \frac{\mathbf{x}}{\lambda}\big)\Big), \quad (21) \\ &= (1/\lambda)^D \mathcal{N}(\boldsymbol{\mu}; \mathbf{x}/\lambda, \boldsymbol{\Sigma}/\lambda^2) \end{aligned}$$

where $\lambda \in \mathbb{R}^+$, $\boldsymbol{\mu}$ denotes a vector with dimension $D$, and $\boldsymbol{\Sigma}$ is a positive semi-definite matrix. Fromt that, and $\beta_t = 1 - \alpha_t$, the following identities follow:

$$
\begin{cases}
\mathcal{N}(\mathbf{x}_t; \sqrt{1 - \beta_t}\mathbf{x}_{t-1}, \beta_t \mathbf{I}) = \alpha_t^{-\frac{D}{2}} \mathcal{N}\Big(\mathbf{x}_{t-1}; \dfrac{\mathbf{x}_t}{\sqrt{\alpha_t}}, \dfrac{1 - \alpha_t}{\alpha_t}\mathbf{I}\Big) \\
\mathcal{N}(\mathbf{x}_{t-1}; \sqrt{\bar{\alpha}_{t-1}}\mathbf{x}_0, (1 - \bar{\alpha}_{t-1})\mathbf{I}) = (\bar{\alpha}_{t-1})^{-\frac{D}{2}} \mathcal{N}\Big(\mathbf{x}_0; \dfrac{\mathbf{x}_{t-1}}{\sqrt{\bar{\alpha}_{t-1}}}, \dfrac{1 - \bar{\alpha}_{t-1}}{\bar{\alpha}_{t-1}}\mathbf{I}\Big)
\end{cases} . \tag{22}
$$

Therefore, we can refomulate Eq. (20) as

$$
q(\cdot) = \frac{(\alpha_t \bar{\alpha}_{t-1})^{-\frac{D}{2}}}{q(\mathbf{x}_t)} \mathcal{N}\Big(\mathbf{x}_{t-1}; \frac{\mathbf{x}_t}{\sqrt{\alpha_t}}, \frac{1 - \alpha_t}{\alpha_t}\mathbf{I}\Big) \int_{\mathbf{x}_0} \mathcal{N}\Big(\mathbf{x}_0; \frac{\mathbf{x}_{t-1}}{\sqrt{\bar{\alpha}_{t-1}}}, \frac{1 - \bar{\alpha}_{t-1}}{\bar{\alpha}_{t-1}}\mathbf{I}\Big) q(\mathbf{x}_0) d\mathbf{x}_0. \tag{23}
$$

Now, we let $q(\mathbf{x}_0)$ be a mixture of Gaussians $q(\mathbf{x}_0) = \sum_{k=1}^{K} w_k \mathcal{N}(\mathbf{x}_0; \boldsymbol{\mu}_k, \boldsymbol{\Sigma}_k)$, where $K$ is the number of Gaussian components, $w_k \in [0, 1]$, $\sum_k w_k = 1$, and vector $\boldsymbol{\mu}_k$ and matrix $\boldsymbol{\Sigma}_k$ respectively denote the mean and covariance of component $k$.

For the the mixture of Gaussians distribution $q(\mathbf{x}_0)$ and by exchanging the operation order of summation $\sum_{k=1}^{K}$ and integral $\int_{\mathbf{x}_0}$, we have

$$
\begin{aligned}
q(\mathbf{x}_{t-1} \mid \mathbf{x}_t) = \sum_{k=1}^{K} \Big[ & \frac{w_k (\alpha_t \bar{\alpha}_{t-1})^{-\frac{D}{2}}}{q(\mathbf{x}_t)} \mathcal{N}\Big(\mathbf{x}_{t-1}; \frac{\mathbf{x}_t}{\sqrt{\alpha_t}}, \frac{1 - \alpha_t}{\alpha_t}\mathbf{I}\Big) \\
& * \int_{\mathbf{x}_0} \mathcal{N}\Big(\mathbf{x}_0; \frac{\mathbf{x}_{t-1}}{\sqrt{\bar{\alpha}_{t-1}}}, \frac{1 - \bar{\alpha}_{t-1}}{\bar{\alpha}_{t-1}}\mathbf{I}\Big) \mathcal{N}\Big(\mathbf{x}_0; \boldsymbol{\mu}_k, \boldsymbol{\Sigma}_k\Big) d\mathbf{x}_0 \Big].
\end{aligned} \tag{24}
$$

A nice property of Gaussian distributions is that the product of two multivariate Gaussians also follows a Gaussian distribution (Ahrendt, 2005). Formally, we have

$$
\begin{aligned}
\mathcal{N}(\mathbf{x}; \boldsymbol{\mu}_1, \boldsymbol{\Sigma}_1) \mathcal{N}(\mathbf{x}; \boldsymbol{\mu}_2, \boldsymbol{\Sigma}_2) = {} & \mathcal{N}(\boldsymbol{\mu}_2; \boldsymbol{\mu}_1, \boldsymbol{\Sigma}_1 + \boldsymbol{\Sigma}_2) \\
& * \mathcal{N}(\mathbf{x}; (\boldsymbol{\Sigma}_1^{-1} + \boldsymbol{\Sigma}_2^{-1})^{-1}(\boldsymbol{\Sigma}_1^{-1}\boldsymbol{\mu}_1 + \boldsymbol{\Sigma}_2^{-1}\boldsymbol{\mu}_2), (\boldsymbol{\Sigma}_1^{-1} + \boldsymbol{\Sigma}_2^{-1})^{-1}),
\end{aligned} \tag{25}
$$

where $\boldsymbol{\mu}_1, \boldsymbol{\mu}_2$ are vectors of the same dimension and $\boldsymbol{\Sigma}_1, \boldsymbol{\Sigma}_2$ are positive-definite matrices. Therefore, the integral part $\int_{\mathbf{x}_0}$ in Eq. (24) can be computed as

$$
\begin{aligned}
& \int_{\mathbf{x}_0} \mathcal{N}\Big(\mathbf{x}_0; \frac{\mathbf{x}_{t-1}}{\sqrt{\bar{\alpha}_{t-1}}}, \frac{1 - \bar{\alpha}_{t-1}}{\bar{\alpha}_{t-1}}\mathbf{I}\Big) \mathcal{N}\Big(\mathbf{x}_0; \boldsymbol{\mu}_k, \boldsymbol{\Sigma}_k\Big) d\mathbf{x}_0 \\
& = \mathcal{N}\Big(\boldsymbol{\mu}_k; \frac{\mathbf{x}_{t-1}}{\sqrt{\bar{\alpha}_{t-1}}}, \frac{1 - \bar{\alpha}_{t-1}}{\bar{\alpha}_{t-1}}\mathbf{I} + \boldsymbol{\Sigma}_k\Big) * \int_{\mathbf{x}_0} \mathcal{N}(\mathbf{x}_0; \cdot, \cdot) d\mathbf{x}_0 \\
& = (\bar{\alpha}_{t-1})^{-\frac{D}{2}} \mathcal{N}(\mathbf{x}_{t-1}; \sqrt{\bar{\alpha}_{t-1}}\boldsymbol{\mu}_k, (1 - \bar{\alpha}_{t-1})\mathbf{I} + \bar{\alpha}_{t-1}\boldsymbol{\Sigma}_k) * 1
\end{aligned} \tag{26}
$$

where the last equation is derived by Eq. (21). With this result, we have

$$
q(\mathbf{x}_{t-1} \mid \mathbf{x}_t) = \sum_{k=1}^{K} \Big[ \frac{w_k \alpha_t^{-\frac{D}{2}}}{q(\mathbf{x}_t)} \mathcal{N}(\cdot) \mathcal{N}\big(\mathbf{x}_{t-1}; \sqrt{\bar{\alpha}_{t-1}}\boldsymbol{\mu}_k, (1 - \bar{\alpha}_{t-1})\mathbf{I} + \bar{\alpha}_{t-1}\boldsymbol{\Sigma}_k\big) \Big], \tag{27}
$$

By applying Eq. (25) and Eq. (21), and $\bar{\alpha}_{t-1}\alpha_t = \bar{\alpha}_t$, the product of two Gaussian distributions in the above equality can be reformulated as

$$
\begin{aligned}
& \mathcal{N}\Big(\mathbf{x}_{t-1}; \frac{\mathbf{x}_t}{\sqrt{\alpha_t}}, \frac{1 - \alpha_t}{\alpha_t}\mathbf{I}\Big) * \mathcal{N}\Big(\mathbf{x}_{t-1}; \sqrt{\bar{\alpha}_{t-1}}\boldsymbol{\mu}_k, (1 - \bar{\alpha}_{t-1})\mathbf{I} + \bar{\alpha}_{t-1}\boldsymbol{\Sigma}_k\Big) \\
& = \alpha_t^{\frac{D}{2}} \mathcal{N}\Big(\mathbf{x}_t; \sqrt{\bar{\alpha}_t}\boldsymbol{\mu}_k, (1 - \bar{\alpha}_t)\mathbf{I} + \bar{\alpha}_t\boldsymbol{\Sigma}_k\Big) \\
& * \mathcal{N}\Big(\mathbf{x}_{t-1}; (\mathbf{I} + \boldsymbol{\Lambda}_k^{-1})^{-1}\frac{\mathbf{x}_t}{\sqrt{\alpha_t}} + (\mathbf{I} + \boldsymbol{\Lambda}_k)^{-1}\sqrt{\bar{\alpha}_{t-1}}\boldsymbol{\mu}_k, \frac{1 - \alpha_t}{\alpha_t}(\mathbf{I} + \boldsymbol{\Lambda}_k^{-1})^{-1}\Big)
\end{aligned} \tag{28}
$$

where matrix $\boldsymbol{\Lambda}_k = (\alpha_t - \bar{\alpha}_t)/(1 - \alpha_t)\mathbf{I} + \bar{\alpha}_t/(1 - \alpha_t)\boldsymbol{\Sigma}_k$. With this result, we have

$$
\begin{cases}
q(\mathbf{x}_{t-1} \mid \mathbf{x}_t) = \displaystyle\sum_{k=1}^{K} w_k' \mathcal{N}(\mathbf{x}_{t-1}; \boldsymbol{\mu}_k', \boldsymbol{\Sigma}_k') \\[2mm]
\quad w_k' = \dfrac{w_k}{q(\mathbf{x}_t)} \mathcal{N}(\mathbf{x}_t; \sqrt{\bar{\alpha}_t}\boldsymbol{\mu}_k, (1 - \bar{\alpha}_t)\mathbf{I} + \bar{\alpha}_t\boldsymbol{\Sigma}_k) \\[2mm]
\quad \boldsymbol{\mu}_k' = (\mathbf{I} + \boldsymbol{\Lambda}_k^{-1})^{-1}\dfrac{\mathbf{x}_t}{\sqrt{\alpha_t}} + (\mathbf{I} + \boldsymbol{\Lambda}_k)^{-1}\sqrt{\bar{\alpha}_{t-1}}\boldsymbol{\mu}_k \\[2mm]
\quad \boldsymbol{\Sigma}_k' = \dfrac{1 - \alpha_t}{\alpha_t}(\mathbf{I} + \boldsymbol{\Lambda}_k^{-1})^{-1}
\end{cases}
, \tag{29}
$$

where $\sum_{k=1}^{K} w_k' = 1$. To conclude, from this equality it follows that posterior probability $p(\mathbf{x}_{t-1} \mid \mathbf{x}_t)$ is also a mixture of Gaussians. Therefore, our proposition holds.

## C  PROOF OF THEOREM 3.1

Let us rewrite metric $\mathcal{M}_t$ as

$$
\begin{aligned}
\mathcal{M}_t &= \inf_{\theta \in \Theta} \left( \int_{\mathbf{x}_t} q(\mathbf{x}_t) \left( \int_{\mathbf{x}_{t-1}} q(\mathbf{x}_{t-1} \mid \mathbf{x}_t) \ln \frac{q(\mathbf{x}_{t-1} \mid \mathbf{x}_t)}{p_\theta(\mathbf{x}_{t-1} \mid \mathbf{x}_t)} d\mathbf{x}_{t-1} \right) d\mathbf{x}_t \right) \\[2mm]
&= \inf_{\theta \in \Theta} \left( \int_{\mathbf{x}_t} q(\mathbf{x}_t) \left( -\mathcal{H}[q(\mathbf{x}_{t-1} \mid \mathbf{x}_t)] + \mathcal{D}_{\mathrm{CE}}[q(\mathbf{x}_{t-1} \mid \mathbf{x}_t), p_\theta(\mathbf{x}_{t-1} \mid \mathbf{x}_t)] \right) d\mathbf{x}_t \right)
\end{aligned}
, \tag{30}
$$

where $\mathcal{H}[\cdot]$ is information entropy (Shannon, 2001):

$$
\mathcal{H}[q(\mathbf{x}_{t-1} \mid \mathbf{x}_t)] = -\int_{\mathbf{x}_{t-1}} q(\mathbf{x}_{t-1} \mid \mathbf{x}_t) \ln q(\mathbf{x}_{t-1} \mid \mathbf{x}_t) d\mathbf{x}_{t-1}, \tag{31}
$$

and $\mathcal{D}_{\mathrm{CE}}[\cdot]$ denotes the cross-entropy (De Boer et al., 2005):

$$
\mathcal{D}_{\mathrm{CE}}[q(\mathbf{x}_{t-1} \mid \mathbf{x}_t), p_\theta(\mathbf{x}_{t-1} \mid \mathbf{x}_t)] = -\int_{\mathbf{x}_{t-1}} q(\mathbf{x}_{t-1} \mid \mathbf{x}_t) \ln p_\theta(\mathbf{x}_{t-1} \mid \mathbf{x}_t) d\mathbf{x}_{t-1}. \tag{32}
$$

Note that the entropy term $\mathcal{H}[\cdot]$ does not involve parameter $\theta$ and can be regarded as a normalization term for adjusting the minimum of $\mathcal{D}_{\mathrm{KL}}[\cdot]$ to 0.

Our goal is to analyze error metric $\mathcal{M}_t$ defined in Eq. (7). Regarding its decomposition derived in Eq. (30), we first focus on cross-entropy $\mathcal{D}_{\mathrm{CE}}[q(\mathbf{x}_{t-1} \mid \mathbf{x}_t), p_\theta(\mathbf{x}_{t-1} \mid \mathbf{x}_t)]$. Suppose $q(\mathbf{x}_0)$ follows a Gaussian mixture, then $q(\mathbf{x}_{t-1} \mid \mathbf{x}_t)$ is also such a distribution as formulated in Eq. (29). Therefore, we can expand the above cross entropy $\mathcal{D}_{\mathrm{CE}}$ as

$$
\begin{aligned}
\mathcal{D}_{\mathrm{CE}}[\cdot] &= -\int_{\mathbf{x}_{t-1}} q(\mathbf{x}_{t-1} \mid \mathbf{x}_t) \ln p_\theta(\mathbf{x}_{t-1} \mid \mathbf{x}_t) d\mathbf{x}_{t-1} \\[2mm]
&= -\int_{\mathbf{x}_{t-1}} \left( \sum_{k=1}^{K} w_k' \mathcal{N}(\mathbf{x}_{t-1}; \boldsymbol{\mu}_k', \boldsymbol{\Sigma}_k') \right) \ln p_\theta(\mathbf{x}_{t-1} \mid \mathbf{x}_t) d\mathbf{x}_{t-1} \\[2mm]
&= \sum_{k=1}^{K} w_k' \mathcal{D}_{\mathrm{CE}}[\mathcal{N}(\mathbf{x}_{t-1}; \boldsymbol{\mu}_k', \boldsymbol{\Sigma}_k'), p_\theta(\mathbf{x}_{t-1} \mid \mathbf{x}_t)] \\[2mm]
&= \sum_{k=1}^{K} w_k' \mathcal{D}_{\mathrm{KL}}[\mathcal{N}(\mathbf{x}_{t-1}; \boldsymbol{\mu}_k', \boldsymbol{\Sigma}_k'), p_\theta(\mathbf{x}_{t-1} \mid \mathbf{x}_t)] + \sum_{k=1}^{K} w_k' \mathcal{H}[\mathcal{N}(\mathbf{x}_{t-1}; \boldsymbol{\mu}_k', \boldsymbol{\Sigma}_k')]
\end{aligned}
. \tag{33}
$$

Suppose we set $\boldsymbol{\Sigma}_k = \delta_k \mathbf{I}, \delta_k > 0$, then we have

$$
\begin{cases}
\boldsymbol{\mu}_k' = \left( \dfrac{1 + (\delta_k - 1)\bar{\alpha}_{t-1}}{1 + (\delta_k - 1)\bar{\alpha}_t} \right) \sqrt{\alpha_t}\mathbf{x}_t + \dfrac{(1 - \alpha_t)\sqrt{\bar{\alpha}_{t-1}}}{1 + (\delta_k - 1)\bar{\alpha}_t}\boldsymbol{\mu}_k \\[3mm]
\boldsymbol{\Sigma}_k' = \left( \dfrac{1 + (\delta_k - 1)\bar{\alpha}_{t-1}}{1 + (\delta_k - 1)\bar{\alpha}_t} \right)(1 - \alpha_t)\mathbf{I}
\end{cases}
. \tag{34}
$$

With this equation, we can simplify entropy sum $\sum_{k=1}^{K} w'_k \mathcal{H}[\cdot]$ as

$$\sum_{k=1}^{K} w'_k \mathcal{H}[\mathcal{N}(\mathbf{x}_{t-1}; \boldsymbol{\mu}'_k, \boldsymbol{\Sigma}'_k)] = \sum_{k=1}^{K} \frac{w'_k}{2} \ln |2\pi \mathrm{e} \boldsymbol{\Sigma}'_k| = \frac{D}{2} \ln(2\pi \mathrm{e}) + \sum_{k=1}^{K} \frac{w'_k}{2} \ln |\boldsymbol{\Sigma}'_k|. \quad (35)$$

Term $\mathcal{D}_{\mathrm{KL}}[\cdot]$ is in fact the KL divergence between two multivariate Gaussians, $\mathcal{N}(\mathbf{x}_{t-1}; \boldsymbol{\mu}'_k, \boldsymbol{\Sigma}'_k)$ and $\mathcal{N}(\mathbf{x}_{t-1}; \boldsymbol{\mu}_\theta(\mathbf{x}_t, t), \sigma_t \mathbf{I})$, which has an analytic form (Zhang et al., 2021):

$$\begin{aligned}
\mathcal{D}_{\mathrm{KL}}[\cdot] &= \frac{1}{2}\Big( \ln \frac{|\sigma_t \mathbf{I}|}{|\boldsymbol{\Sigma}'_k|} - D + \frac{1}{\sigma_t}\|\boldsymbol{\mu}'_k - \boldsymbol{\mu}_\theta(\mathbf{x}_t, t)\|^2 + \mathrm{Tr}\{(\sigma_t \mathbf{I})^{-1} \boldsymbol{\Sigma}'_k\} \Big) \\
&= \frac{1}{2}\Big( D \ln \sigma_t - \ln |\boldsymbol{\Sigma}'_k| - D \Big) + \frac{1}{2\sigma_t}\|\boldsymbol{\mu}'_k - \boldsymbol{\mu}_\theta(\mathbf{x}_t, t)\|^2 + \frac{1-\alpha_t}{2\sigma_t}\frac{1+(\delta_k-1)\bar{\alpha}_{t-1}}{1+(\delta_k-1)\bar{\alpha}_t} D
\end{aligned} \quad (36)$$

With the above two equalities and the fact that $\bar{\alpha}_{t-1} > \bar{\alpha}_t$ because $\alpha_t < 1$, we reduce term $\mathcal{D}_{\mathrm{CE}}[q(\mathbf{x}_{t-1} \mid \mathbf{x}_t), p_\theta(\mathbf{x}_{t-1} \mid \mathbf{x}_t)]$ as

$$\mathcal{D}_{\mathrm{CE}}[\cdot] > \frac{1}{2\sigma_t} \sum_{k=1}^{K} w'_k \|\boldsymbol{\mu}'_k - \boldsymbol{\mu}_\theta(\mathbf{x}_t, t)\|^2 + \frac{D}{2} \ln(2\pi\sigma_t) + \frac{1-\alpha_t}{2\sigma_t} D. \quad (37)$$

Since entropy $\mathcal{H}[q(\mathbf{x}_{t-1} \mid \mathbf{x}_t)]$ does not involve model parameter $\theta$, the variation of error metric $\mathcal{M}_t$ is from cross-entropy $\mathcal{D}_{\mathrm{CE}}[\cdot]$, more specifically, sum $\sum_{k=1}^{K}$. Let's focus on how this term contributes to error metric $\mathcal{M}_t$ as formulated in Eq. (7):

$$\mathcal{I}_{\mathrm{CE}} = \int_{\mathbf{x}_t} q(\mathbf{x}) \sum_{k=1}^{K} w'_k \|\boldsymbol{\mu}'_k - \boldsymbol{\mu}_\theta(\mathbf{x}_t, t)\|^2 d\mathbf{x}_t = \sum_{k=1}^{K} \Big( \int_{\mathbf{x}_t} w'_k q(\mathbf{x}) \|\boldsymbol{\mu}'_k - \boldsymbol{\mu}_\theta(\mathbf{x}_t, t)\|^2 d\mathbf{x}_t \Big). \quad (38)$$

Considering that Eq. (29) and $\boldsymbol{\Sigma}_k$ has been set as $\delta_k \mathbf{I}$, we have

$$\begin{aligned}
\mathcal{I}_{\mathrm{CE}} &= \sum_{k=1}^{K} \Big( \int_{\mathbf{x}_t} w_k \mathcal{N}\Big(\mathbf{x}_t; \sqrt{\bar{\alpha}_t}\boldsymbol{\mu}_k, (1+(\delta_k-1)\bar{\alpha}_t)\mathbf{I}\Big) \Big\|\boldsymbol{\mu}'_k - \boldsymbol{\mu}_\theta(\mathbf{x}_t, t)\Big\|^2 d\mathbf{x}_t \Big) \\
&= \int_{\mathbf{x}_t} \mathcal{N}(\cdot)\Big( \sum_{k=1}^{K} w_k \Big\|\Big(\frac{(1-\alpha_t)\sqrt{\bar{\alpha}_{t-1}}}{1+(\delta_k-1)\bar{\alpha}_t}\Big)\boldsymbol{\mu}_k - \Big(\boldsymbol{\mu}_\theta(\mathbf{x}_t, t) - (\cdot)\sqrt{\alpha_t}\mathbf{x}_t\Big)\Big\|^2\Big) d\mathbf{x}_t
\end{aligned} \quad (39)$$

Sum $\sum_{k=1}^{K} w_k \|\cdot\|^2$ is essentially a problem called weighted least squares (Rousseeuw & Leroy, 2005) for model $\boldsymbol{\mu}_\theta(\mathbf{x}_t, t) - (\cdot)\sqrt{\alpha_t}\mathbf{x}_t$, which achieves a minimum error when the model is $\sum_{k=1}^{K} w_k(\cdot)\boldsymbol{\mu}_k$. For convenience, we suppose $\sum_{k=1}^{K} w_k \boldsymbol{\mu}_k/(1+(\delta_k-1)\bar{\alpha}_t) = \mathbf{0}$ and we have

$$\mathcal{I}_{\mathrm{CE}} \geqslant \Big( \int_{\mathbf{x}_t} \mathcal{N}(\cdot) d\mathbf{x}_t \Big) \Big( \sum_{k=1}^{K} w_k \Big\|(\cdot)\boldsymbol{\mu}_k\Big\|^2 \Big) = (1-\alpha_t)^2 \bar{\alpha}_{t-1} \sum_{k=1}^{K} w_k \Big\|\frac{\boldsymbol{\mu}_k}{1+(\delta_k-1)\bar{\alpha}_t}\Big\|^2. \quad (40)$$

Term $\mathcal{H}[q(\mathbf{x}_{t-1} \mid \mathbf{x}_t)]$ is in fact the differential entropy of a Gaussian mixture. Considering our previous setup and its upper bound provided by (Huber et al., 2008), we have

$$\begin{aligned}
\mathcal{H}[\cdot] &\leqslant \sum_{k=1}^{K} w'_k \Big( -\ln w'_k + \frac{1}{2}\ln\Big((2\pi \mathrm{e})^D \Big|\frac{1+(\delta_k-1)\bar{\alpha}_{t-1}}{1+(\delta_k-1)\bar{\alpha}_t}(1-\alpha_t)\mathbf{I}\Big|\Big)\Big) \\
&< \frac{D}{2}\ln\Big(\frac{2\pi \mathrm{e}}{\alpha_t}(1-\alpha_t)\Big) - \sum_{k=1}^{K} w'_k \ln w'_k \leqslant \frac{D}{2}\ln\Big(2\pi \mathrm{e}\Big(\frac{1}{\alpha_t}-1\Big)\Big) + \ln K
\end{aligned}, \quad (41)$$

where the second inequality holds since $(1+x)/(1+xy) < 1/y, \forall x \in \mathbb{R}^+, y \in (0,1)$ and the last inequality is obtained by regarding term $-\sum_{k=1}^{K}$ as the entropy of discrete variables $[w'_1, w'_2, \cdots, w'_K]$. Therefore, its contribution to error metric $\mathcal{M}_t$ is

$$\mathcal{I}_{\mathrm{Ent}} = \int_{\mathbf{x}_t} q(\mathbf{x}_t)(-\mathcal{H}[q(\mathbf{x}_{t-1} \mid \mathbf{x}_t)]) d\mathbf{x}_t \geqslant -\frac{D}{2}\ln\Big(\frac{2\pi \mathrm{e}}{\alpha_t}(1-\alpha_t)\Big) - \ln K. \quad (42)$$

Combining this inequality with Eq. (37) and Eq. (40), we have

$$\mathcal{M}_t > \frac{(1-\alpha_t)^2 \bar{\alpha}_{t-1}}{2\sigma_t} \sum_{k=1}^{K} w_k \Big\|\frac{\boldsymbol{\mu}_k}{1+(\delta_k-1)\bar{\alpha}_t}\Big\|^2 - \ln K + \frac{D}{2}\Big( \ln \frac{\sigma_t \alpha_t}{1-\alpha_t} + \frac{1-\alpha_t}{\sigma_t} - 1 \Big). \quad (43)$$

with constraint $\sum_{k=1}^{K} w_k \boldsymbol{\mu}_k/(1 + (\delta_k - 1)\bar{\alpha}_t) = \mathbf{0}$. Since $w_k > 0, 1 \leqslant k \leqslant K$, there exists a group of non-zero vectors $[\boldsymbol{\mu}_1, \boldsymbol{\mu}_2, \cdots, \boldsymbol{\mu}_K]$ satisfying this linear equation, corresponds to a Gaussian mixture $p(\mathbf{x}_0)$. With this result, we can always find another group of solution $[\lambda\boldsymbol{\mu}_1, \lambda\boldsymbol{\mu}_2, \cdots, \lambda\boldsymbol{\mu}_K]$ for $\lambda \in \mathbb{R}$, which corresponds to a new mixture of Gaussians. By increasing the value of $\lambda$, the first term of this inequality can be arbitrarily and uniformly large in terms of iteration $t$.

## D  PROOF OF THEOREM 3.2

Due to the first-order markov property of the forward and backward processes and the fact $q(\mathbf{x}_T) = p_\theta(\mathbf{x}_T) = \mathcal{N}(\mathbf{0}, \mathbf{I}), T \to \infty$, we first have

$$
\begin{aligned}
\mathcal{D}_{\mathrm{KL}}[\cdot] = \mathbb{E}_{\mathbf{x}_{0:T} \sim q(\mathbf{x}_{0:T})}\Big[ \ln \frac{q(\mathbf{x}_{0:T})}{p_\theta(\mathbf{x}_{0:T})} \Big] &= \mathbb{E}_q\Big[ \ln \frac{q(\mathbf{x}_T)\prod_{t=T}^{1} q(\mathbf{x}_{t-1} \mid \mathbf{x}_t)}{p_\theta(\mathbf{x}_T)\prod_{t=T}^{1} p_\theta(\mathbf{x}_{t-1} \mid \mathbf{x}_t)} \Big], \\
= \mathbb{E}_q\Big[ \sum_{t=1}^{T} \ln \frac{q(\mathbf{x}_{t-1} \mid \mathbf{x}_t)}{p_\theta(\mathbf{x}_{t-1} \mid \mathbf{x}_t)} \Big] &= \sum_{t=1}^{T} E_{\mathbf{x}_t}\Big[ \mathcal{D}_{\mathrm{KL}}[q(\mathbf{x}_{t-1} \mid \mathbf{x}_t), p_\theta(\mathbf{x}_{t-1} \mid \mathbf{x}_t)] \Big]
\end{aligned}
\tag{44}
$$

where the last equality holds because of the following derivation:

$$
\begin{aligned}
\mathbb{E}_q\Big[ \ln \frac{q(\mathbf{x}_{t-1} \mid \mathbf{x}_t)}{p_\theta(\mathbf{x}_{t-1} \mid \mathbf{x}_t)} \Big] &= \int_{\mathbf{x}_{0:T}} q(\mathbf{x}_{0:T}) \ln \frac{q(\mathbf{x}_{t-1} \mid \mathbf{x}_t)}{p_\theta(\mathbf{x}_{t-1} \mid \mathbf{x}_t)} d\mathbf{x}_{0:T} \\
&= \int_{\mathbf{x}_{t-1}} q(\mathbf{x}_t)\Big( \int_{\mathbf{x}_t} q(\mathbf{x}_{t-1} \mid \mathbf{x}_t) \ln \frac{q(\mathbf{x}_{t-1} \mid \mathbf{x}_t)}{p_\theta(\mathbf{x}_{t-1} \mid \mathbf{x}_t)} d\mathbf{x}_{t-1} \Big) d\mathbf{x}_t \\
&= E_{\mathbf{x}_t \sim q(\mathbf{x}_t)}\Big[ \mathcal{D}_{\mathrm{KL}}[q(\mathbf{x}_{t-1} \mid \mathbf{x}_t), p_\theta(\mathbf{x}_{t-1} \mid \mathbf{x}_t)] \Big].
\end{aligned}
\tag{45}
$$

Based on Theorem 3.1, then we can infer that there is a continuous data distribution $q(\mathbf{x}_0)$ such that the inequality $\mathcal{M}_t > (N + 1)/T$ holds for $t \in [1, T]$. For this distribution, we have

$$
\mathcal{D}_{\mathrm{KL}}[\cdot] \geqslant \sum_{t=1}^{T} \inf \Big( E_{\mathbf{x}_t}\Big[ \mathcal{D}_{\mathrm{KL}}[q(\mathbf{x}_{t-1} \mid \mathbf{x}_t), p_\theta(\mathbf{x}_{t-1} \mid \mathbf{x}_t)] \Big] \Big) = \sum_{t=1}^{T} M_t > N + 1.
\tag{46}
$$

Finally, we get $\mathcal{E} = \inf(\mathcal{D}_{\mathrm{KL}}[\cdot]) \geqslant N + 1 > N$ for the data distribution $q(\mathbf{x}_0)$.

## E  PROOF OF THEOREM 4.1

We split the proof into two parts: one for $\mathcal{M}_t, t \in [1, T]$ and the other for $\mathcal{E}$.

**Zero local denoising errors.**   For convenience, we denote integral $\int_{\mathbf{x}_t} q(\mathbf{x}_t)\mathcal{D}_{\mathrm{KL}}[\cdot]d\mathbf{x}_t$ in the definition of error measure $\mathcal{M}_t$ as $\mathcal{M}_t(\bar{\theta})$. Immediately, we have $\mathcal{M}_t = \inf_{\bar{\theta} \in \bar{\Theta}} \mathcal{M}_t(\bar{\theta})$. With this equality, it suffices to prove two assertions: $\mathcal{M}_t(\bar{\theta}) \geqslant 0, \forall\bar{\theta} \in \Theta$ and $\exists\bar{\theta} \in \bar{\Theta} : \mathcal{M}_t(\bar{\theta}) = 0$.

The first assertion is trivially true since KL divergence $\mathcal{D}_{\mathrm{KL}}$ is always non-negative. For the second assertion, we introduce two lemmas: 1) The assertion is true for the mixture model $p_\theta^{\mathrm{mixture}}(\mathbf{x}_{t-1} \mid \mathbf{x}_t)$; 2) Any mixture model can be represented by its soft version $p_\theta^{\mathrm{SMD}}(\mathbf{x}_{t-1} \mid \mathbf{x}_t)$. If we can prove the two lemma, it is sufficient to say that the second assertion also holds for SMD.

We prove the first lemma by construction. According to Proposition 3.1, the inverse forward probability $q(\mathbf{x}_{t-1} \mid \mathbf{x}_t)$ is also a Gaussian mixture as formulated in Eq. (29). By selecting a proper number $K$, the mixture model $p_\theta^{\mathrm{mixture}}(\mathbf{x}_{t-1} \mid \mathbf{x}_t)$ defined in Eq. (9) will be of the same distribution family as its reference $q(\mathbf{x}_{t-1} \mid \mathbf{x}_t)$, which only differ in the configuration of different mixture components. Based on Eq. (29), we can specifically set parameter $\theta = \bigcup_{1 \leqslant k \leqslant K} \theta_k$ as

$$
\begin{cases}
z_{\theta_k}(\mathbf{x}_t, t) \propto w_k \mathcal{N}(\mathbf{x}_t; \sqrt{\bar{\alpha}_t}\boldsymbol{\mu}_k, (1 - \bar{\alpha}_t)\mathbf{I} + \bar{\alpha}_t\boldsymbol{\Sigma}_k) \\[2mm]
\boldsymbol{\mu}_{\theta_k}(\mathbf{x}_t, t) = (\mathbf{I} + \boldsymbol{\Lambda}_k^{-1})^{-1}\frac{\mathbf{x}_t}{\sqrt{\alpha_t}} + (\mathbf{I} + \boldsymbol{\Lambda}_k)^{-1}\sqrt{\bar{\alpha}_{t-1}}\boldsymbol{\mu}_k \\[2mm]
\boldsymbol{\Sigma}_{\theta_k}(\mathbf{x}_t, t) = \frac{1 - \alpha_t}{\alpha_t}(\mathbf{I} + \boldsymbol{\Lambda}_k^{-1})^{-1} \\[2mm]
\boldsymbol{\Lambda}_k = \frac{\alpha_t - \bar{\alpha}_t}{1 - \alpha_t}\mathbf{I} + \frac{\bar{\alpha}_t}{1 - \alpha_t}\boldsymbol{\Sigma}_k
\end{cases},
\tag{47}
$$

such that the backward probability $p_\theta^{\text{mixture}}(\mathbf{x}_{t-1} \mid \mathbf{x}_t)$ is the same as its reference $q(\mathbf{x}_{t-1} \mid \mathbf{x}_t)$ and thus $\mathcal{D}_{\text{KL}}[q(\mathbf{x}_{t-1} \mid \mathbf{x}_t), p_\theta^{\text{mixture}}(\mathbf{x}_{t-1} \mid \mathbf{x}_t)]$ by definition is 0. In this sense, we also have $\mathcal{M}_t(\theta) = 0$, which exactly proves the first lemma.

We also prove the second lemma by construction. Given any mixture model $p_\theta^{\text{mixture}}(\mathbf{x}_{t-1} \mid \mathbf{x}_t)$ as defined in Eq. (9), we divide the space $\mathbb{R}^L$ (where $L$ is the vector dimension of variable $\mathbf{z}_t$) into $K$ disjoint subsets $\{\mathcal{Z}_{t,1}, \mathcal{Z}_{t,2}, \cdots, \mathcal{Z}_{t,K}\}$ such that:

$$\int_{\mathbf{z}_t \in \mathcal{Z}_{t,k}} p_{\bar{\theta}}^{\text{SMD}}(\mathbf{z}_t \mid \mathbf{x}_t) d\mathbf{z}_t = z_{\theta_k}(\mathbf{x}_t, t), \quad \theta_k = f_\phi(\mathbf{z}_t, t), \forall \mathbf{z}_t \in \mathcal{Z}_{t,k}, \tag{48}$$

where $k \in \{1, ..., K\}$. The first equality can be true for any continuous density $p_{\bar{\theta}}^{\text{SMD}}$ and the second one can be implemented by a simple step function. By setting $\theta = \varnothing$, we have

$$
\begin{aligned}
p_{\bar{\theta}}^{\text{SMD}}(\mathbf{x}_{t-1} \mid \mathbf{x}_t) &= \int_{\mathbf{z}_t} p_{\bar{\theta}}^{\text{SMD}}(\mathbf{z}_t \mid \mathbf{x}_t) \mathcal{N}(\mathbf{x}_{t-1}; \boldsymbol{\mu}_{\theta, f_\phi(\mathbf{z}_t, t)}(\mathbf{x}_t, t), \boldsymbol{\Sigma}_{\theta, f_\phi(\mathbf{z}_t, t)}(\mathbf{x}_t, t)) d\mathbf{z}_t \\
&= \sum_{k=1}^{K} \Big( \int_{\mathbf{z}_t \in \mathcal{Z}_{t,k}} p_{\bar{\theta}}^{\text{SMD}}(\mathbf{z}_t \mid \mathbf{x}_t) \mathcal{N}\big(\mathbf{x}_{t-1}; \boldsymbol{\mu}_{f_\phi(\cdot)}(\mathbf{x}_t, t), \boldsymbol{\Sigma}_{f_\phi(\cdot)}(\mathbf{x}_t, t)\big) d\mathbf{z}_t \Big) \\
&= \sum_{k=1}^{K} \Big( \mathcal{N}\big(\mathbf{x}_{t-1}; \boldsymbol{\mu}_{\theta_k}(\mathbf{x}_t, t), \boldsymbol{\Sigma}_{\theta_k}(\mathbf{x}_t, t)\big) \int_{\mathbf{z}_t \in \mathcal{Z}_{t,k}} p_{\bar{\theta}}^{\text{SMD}}(\mathbf{z}_t \mid \mathbf{x}_t) d\mathbf{z}_t \Big) \\
&= \sum_{k=1}^{K} \Big( \mathcal{N}(\mathbf{x}_{t-1}; \boldsymbol{\mu}_{\theta_k}(\mathbf{x}_t, t), \boldsymbol{\Sigma}_{\theta_k}(\mathbf{x}_t, t)) z_{\theta_k}(\mathbf{x}_t, t) \Big) = p_\theta^{\text{mixture}}(\mathbf{x}_{t-1} \mid \mathbf{x}_t)
\end{aligned}
\tag{49}
$$

which actually proves the second lemma.

**Zero global denoising error.** We can see from above that there is always a properly parameterized backward probability $p_{\bar{\theta}}^{\text{SMD}}$ for any Gaussian mixture $q(\mathbf{x}_0)$ such that $q(\mathbf{x}_{t-1} \mid \mathbf{x}_t) = p_{\bar{\theta}}^{\text{SMD}}(\mathbf{x}_{t-1} \mid \mathbf{x}_t), \forall t \in [1, T]$. Considering $q(\mathbf{x}_T) = p_{\bar{\theta}}^{\text{SMD}}(\mathbf{x}_T)$, we have

$$p_{\bar{\theta}}^{\text{SMD}}(\mathbf{x}_{T-1}, \mathbf{x}_T) = p_{\bar{\theta}}^{SMD}(\mathbf{x}_T) p_{\bar{\theta}}^{\text{SMD}}(\mathbf{x}_{T-1} \mid \mathbf{x}_T) = q(\mathbf{x}_T) q(\cdot) = q(\mathbf{x}_{T-1}, \mathbf{x}_T). \tag{50}$$

Immediately, we can get $q(\mathbf{x}_{T-1}) = p_{\bar{\theta}}^{\text{SMD}}(\mathbf{x}_{T-1})$ since

$$p_{\bar{\theta}}^{\text{SMD}}(\mathbf{x}_{T-1}) = \int_{\mathbf{x}_T} p_{\bar{\theta}}^{\text{SMD}}(\mathbf{x}_{T-1}, \mathbf{x}_T) \mathbf{x}_T = \int_{\mathbf{x}_T} q(\mathbf{x}_{T-1}, \mathbf{x}_T) \mathbf{x}_T = q(\mathbf{x}_{T-1}). \tag{51}$$

With the above results, we can further prove that $p_{\bar{\theta}}^{\text{SMD}}(\mathbf{x}_{T-2}, \mathbf{x}_{T-1}, \mathbf{x}_T) = q(\mathbf{x}_{T-2}, \mathbf{x}_{T-1}, \mathbf{x}_T)$ and $p_{\bar{\theta}}^{\text{SMD}}(\mathbf{x}_{T-2}) = q(\mathbf{x}_{T-2})$. By iterating this process for the subscript $t$ from $T$ to 1, we will finally have $p_{\bar{\theta}}(\mathbf{x}_{0:T}) = q(\mathbf{x}_{0:T})$ such that $\mathcal{E} = 0$.

## F  PROOF OF PROPOSITION 4.1

While we have introduced a new family of backward probability $p_\theta^{\text{SMD}}(\mathbf{x}_{t-1} \mid \mathbf{x}_t)$ in Eq. (10), upper bound $\mathcal{L} = \sum_{t=0}^{T} \mathcal{L}_t$ defined in Eq. (3) is still valid for deriving the loss function. To avoid confusion, we add a superscript SMD to new loss terms. An immediate conclusion is that $\mathcal{L}_T^{\text{SMD}} = 0$ because $p(\mathbf{x}_t)$ by definition is a standard Gaussian and $q(\mathbf{x}_T \mid \mathbf{x}_0)$ also well approximates this distribution for large $T$. Therefore, the focus of this proof is on terms of KL divergence $\mathcal{L}_{t-1}^{\text{SMD}}, 1 < t \leqslant T$ and negative log-likelihood $\mathcal{L}_0^{\text{SMD}}$.

Based on the fact that $q(\mathbf{x}_{t-1} \mid \mathbf{x}_t, \mathbf{x}_0)$ has a closed-form solution:

$$q(\mathbf{x}_{t-1} \mid \mathbf{x}_t, \mathbf{x}_0) = \mathcal{N}(\mathbf{x}_{t-1}; \widetilde{\boldsymbol{\mu}}_t(\mathbf{x}_t, \mathbf{x}_0), \widetilde{\beta}_t \mathbf{I}), \tag{52}$$

where mean $\widetilde{\boldsymbol{\mu}}_t(\mathbf{x}_t, \mathbf{x}_0)$ and variance $\widetilde{\beta}_t$ are respectively defined as

$$\widetilde{\boldsymbol{\mu}}_t(\mathbf{x}_t, \mathbf{x}_0) = \frac{\sqrt{\bar{\alpha}_{t-1}} \beta_t}{1 - \bar{\alpha}_t} \mathbf{x}_0 + \frac{\sqrt{\alpha_t}(1 - \bar{\alpha}_{t-1})}{1 - \bar{\alpha}_t} \mathbf{x}_t, \quad \widetilde{\beta}_t = \frac{1 - \bar{\alpha}_{t-1}}{1 - \bar{\alpha}_t} \beta_t, \tag{53}$$

we expand term $\mathcal{L}_{t-1}^{\text{SMD}} = \mathbb{E}_q[D_{\text{KL}}(q(\mathbf{x}_{t-1} \mid \mathbf{x}_t, \mathbf{x}_0) \parallel p_{\bar{\theta}}^{\text{SMD}}(\mathbf{x}_{t-1} \mid \mathbf{x}_t))]$ as

$$
\begin{aligned}
\mathcal{L}_{t-1}^{\text{SMD}} &= \mathbb{E}_{\mathbf{x}_0, \mathbf{x}_t \sim q(\mathbf{x}_0) q(\mathbf{x}_t | \mathbf{x}_0)} \Big[ \int_{\mathbf{x}_{t-1}} q(\mathbf{x}_{t-1} \mid \mathbf{x}_t, \mathbf{x}_0) \ln \frac{q(\mathbf{x}_{t-1} \mid \mathbf{x}_t, \mathbf{x}_0)}{p_{\bar{\theta}}^{\text{SMD}}(\mathbf{x}_{t-1} \mid \mathbf{x}_t)} d\mathbf{x}_{t-1} \Big] \\
&= \mathbb{E}_q \Big[ - \mathcal{H}\big[ q(\mathbf{x}_{t-1} \mid \mathbf{x}_t, \mathbf{x}_0) \big] + \mathcal{D}_{\text{CE}}\big[ q(\mathbf{x}_{t-1} \mid \mathbf{x}_t, \mathbf{x}_0), p_{\bar{\theta}}^{\text{SMD}}(\mathbf{x}_{t-1} \mid \mathbf{x}_t) \big] \Big]
\end{aligned}
\tag{54}
$$

Considering our new definition of backward probability $p_{\bar{\theta}}^{\text{SMD}}(\mathbf{x}_{t-1} \mid \mathbf{x}_t)$ in Eq. (10) and applying Jensen's inequality, we can infer

$$
\begin{aligned}
\mathcal{D}_{\text{CE}}[\cdot] &= -\mathbb{E}_{\mathbf{x}_{t-1} \sim q(\mathbf{x}_{t-1} | \mathbf{x}_t, \mathbf{x}_0)} \Big[ \ln \int_{\mathbf{z}_t} p_{\bar{\theta}}^{\text{SMD}}(\mathbf{z}_t \mid \mathbf{x}_t) p_{\bar{\theta}}^{\text{SMD}}(\mathbf{x}_{t-1} \mid \mathbf{x}_t, \mathbf{z}_t) d\mathbf{z}_t \Big] \\
&= -\mathbb{E}_{\mathbf{x}_{t-1} \sim q(\mathbf{x}_{t-1} | \mathbf{x}_t, \mathbf{x}_0)} \Big[ \ln \mathbb{E}_{\mathbf{z}_t \sim p_{\bar{\theta}}^{\text{SMD}}(\mathbf{z}_t | \mathbf{x}_t)} [ p_{\bar{\theta}}^{\text{SMD}}(\mathbf{x}_{t-1} \mid \mathbf{x}_t, \mathbf{z}_t) d\mathbf{z}_t ] \Big] \\
&\leqslant -\mathbb{E}_{\mathbf{x}_{t-1} \sim q(\mathbf{x}_{t-1} | \mathbf{x}_t, \mathbf{x}_0)} \Big[ \mathbb{E}_{\mathbf{z}_t \sim p_{\bar{\theta}}^{\text{SMD}}(\mathbf{z}_t | \mathbf{x}_t)} [ \ln p_{\bar{\theta}}^{\text{SMD}}(\mathbf{x}_{t-1} \mid \mathbf{x}_t, \mathbf{z}_t) d\mathbf{z}_t ] \Big] \\
&= \mathbb{E}_{\mathbf{z}_t \sim p_{\bar{\theta}}^{\text{SMD}}(\mathbf{z}_t | \mathbf{x}_t)} \Big[ - \int_{\mathbf{x}_{t-1}} q(\mathbf{x}_{t-1} \mid \mathbf{x}_t, \mathbf{x}_0) \ln p_{\bar{\theta}}^{\text{SMD}}(\mathbf{x}_{t-1} \mid \mathbf{x}_t, \mathbf{z}_t) d\mathbf{x}_{t-1} \Big] \\
&= \mathbb{E}_{\mathbf{z}_t \sim p_{\bar{\theta}}^{\text{SMD}}(\mathbf{z}_t | \mathbf{x}_t)} \Big[ \mathcal{D}_{\text{CE}}\big[ q(\mathbf{x}_{t-1} \mid \mathbf{x}_t, \mathbf{x}_0), p_{\bar{\theta}}^{\text{SMD}}(\mathbf{x}_{t-1} \mid \mathbf{x}_t, \mathbf{z}_t) \big] \Big]
\end{aligned}
\tag{55}
$$

Combining the above two equations, we have

$$
\begin{aligned}
\mathcal{L}_{t-1}^{\text{SMD}} &\leqslant \mathbb{E}_q \Big[ - \mathcal{H}[q(\mathbf{x}_{t-1} \mid \mathbf{x}_t, \mathbf{x}_0)] + \mathbb{E}_{\mathbf{z}_t}[\mathcal{D}_{\text{CE}}[q(\mathbf{x}_{t-1} \mid \mathbf{x}_t, \mathbf{x}_0), p_{\bar{\theta}}^{\text{SMD}}(\mathbf{x}_{t-1} \mid \mathbf{x}_t, \mathbf{z}_t)]] \Big] \\
&= \mathbb{E}_{q, \mathbf{z}_t} \Big[ - \mathcal{H}[q(\mathbf{x}_{t-1} \mid \mathbf{x}_t, \mathbf{x}_0)] + \mathcal{D}_{\text{CE}}[q(\mathbf{x}_{t-1} \mid \mathbf{x}_t, \mathbf{x}_0), p_{\bar{\theta}}^{\text{SMD}}(\mathbf{x}_{t-1} \mid \mathbf{x}_t, \mathbf{z}_t)] \Big] \\
&= \mathbb{E}_{\mathbf{z}_t} \Big[ \mathbb{E}_{\mathbf{x}_0, \mathbf{x}_t \sim q(\mathbf{x}_0) q(\mathbf{x}_t | \mathbf{x}_0)} [\mathcal{D}_{\text{KL}}[q(\mathbf{x}_{t-1} \mid \mathbf{x}_t, \mathbf{x}_0), p_{\bar{\theta}}^{\text{SMD}}(\mathbf{x}_{t-1} \mid \mathbf{x}_t, \mathbf{z}_t)]] \Big]
\end{aligned}
\tag{56}
$$

Considering $\mathbf{z}_t = g_\varphi(\boldsymbol{\eta}, \mathbf{x}_t, t)$ and applying the law of the unconscious statistician (LOTUS) (Rezende & Mohamed, 2015), we can simplify the above inequality as

$$
\mathcal{L}_{t-1}^{\text{SMD}} \leqslant \mathbb{E}_{\boldsymbol{\eta} \sim \mathcal{N}(\mathbf{0}, \mathbf{I})} \big[ \mathbb{E}_q [\mathcal{D}_{\text{KL}}[q(\mathbf{x}_{t-1} \mid \mathbf{x}_t, \mathbf{x}_0), p_{\bar{\theta}}^{\text{SMD}}(\mathbf{x}_{t-1} \mid \mathbf{x}_t, g_\xi(\boldsymbol{\eta}, \mathbf{x}_t, t))]] \big].
\tag{57}
$$

The inner term of expectation $\mathbb{E}_{\boldsymbol{\eta} \sim \mathcal{N}(\mathbf{0}, \mathbf{I})}[\cdot]$ is essentially the same as the old definition of $\mathcal{L}_t^{\text{SMD}}$ in Eq. (3), except that term $p_{\bar{\theta}}(\cdot)$ is additionally conditional on $\mathbf{z}_t$. Hence, we follow the procedure of DDPM Ho et al. (2020) to reduce it. The result is given without proving:

$$
\begin{cases}
\mathcal{L}_{t-1}^{\text{SMD}} \leqslant C_t + \mathbb{E}_{\boldsymbol{\eta}, \boldsymbol{\epsilon}, \mathbf{x}_0} \Big[ \dfrac{\beta_t^2}{2\sigma_t \alpha_t (1 - \bar{\alpha}_t)} \| \boldsymbol{\epsilon} - \boldsymbol{\epsilon}_{\theta, f_\phi(\cdot)}(\sqrt{\bar{\alpha}_t} \mathbf{x}_0 + \sqrt{1 - \bar{\alpha}_t} \boldsymbol{\epsilon}, t) \|^2 \Big], \\
f_\phi(\cdot) = f_\phi(g_\xi(\boldsymbol{\eta}, \sqrt{\bar{\alpha}_t} \mathbf{x}_0 + \sqrt{1 - \bar{\alpha}_t} \boldsymbol{\epsilon}, t), t)
\end{cases}
\tag{58}
$$

where $C_t$ is a constant, $\boldsymbol{\eta}, \boldsymbol{\epsilon} \sim \mathcal{N}(\mathbf{0}, \mathbf{I})$, and parameters $\theta, \phi, \xi$ are learnable.

For the negative log-likelihood $\mathcal{L}_0^{\text{SMD}} = \mathbb{E}_q[-\ln p_{\bar{\theta}}^{\text{SMD}}(\mathbf{x}_0 \mid \mathbf{x}_1)]$, we expand it as

$$
\mathcal{L}_0^{\text{SMD}} = \mathbb{E}_{\mathbf{x}_0, \mathbf{x}_1 \sim q(\mathbf{x}_0) q(\mathbf{x}_1 | \mathbf{x}_0)} \Big[ -\ln \Big( \int_{\mathbf{z}_1} p_{\bar{\theta}}^{\text{SMD}}(\mathbf{z}_1 \mid \mathbf{x}_1) p_{\bar{\theta}}^{\text{SMD}}(\mathbf{x}_0 \mid \mathbf{x}_1, \mathbf{z}_1) d\mathbf{z}_1 \Big) \Big].
\tag{59}
$$

By applying Jensen's inequality, we have

$$
\begin{aligned}
\mathcal{L}_0^{\text{SMD}} &\leqslant \mathbb{E}_{\mathbf{x}_0, \mathbf{x}_1} \Big[ - \int_{\mathbf{z}_1} p_{\bar{\theta}}^{\text{SMD}}(\mathbf{z}_1 \mid \mathbf{x}_1) \ln p_{\bar{\theta}}^{\text{SMD}}(\mathbf{x}_0 \mid \mathbf{x}_1, \mathbf{z}_1) d\mathbf{z}_1 \Big] \\
&= \mathbb{E}_{\mathbf{x}_0, \mathbf{x}_1} \Big[ \mathbb{E}_{\mathbf{z}_1 \sim p_{\bar{\theta}}^{\text{SMD}}(\mathbf{z}_1 | \mathbf{x}_1)} [-\ln p_{\bar{\theta}}(\mathbf{x}_0 \mid \mathbf{x}_1, \mathbf{z}_1)] \Big] \\
&= C_1 + \mathbb{E}_{\mathbf{z}_1 \sim p_{\bar{\theta}}^{\text{SMD}}(\mathbf{z}_1 | \mathbf{x}_1)} \Big[ \mathbb{E}_{\mathbf{x}_0, \mathbf{x}_1} \Big[ \frac{1}{2\sigma_1} \| \mathbf{x}_0 - \boldsymbol{\mu}_{\theta, f_\phi(\mathbf{z}_1, t)}(\mathbf{x}_1, 1) \|^2 \Big] \Big]
\end{aligned}
\tag{60}
$$

where $C_1$ is a constant that does not involve with the model parameter $\bar{\theta} = \theta \bigcup \phi \bigcup \xi$. Considering Eq. (1) and Eq. (12), we can convert this inequality into

$$
\begin{aligned}
\mathcal{L}_0^{\text{SMD}} &\leqslant C_1 + \mathbb{E}_{\mathbf{z}_1 \sim p_{\bar{\theta}}^{\text{SMD}}(\mathbf{z}_1 | \mathbf{x}_1)} \Big[ \mathbb{E}_{\mathbf{x}_0, \boldsymbol{\epsilon}} \Big[ \frac{\beta_1^2}{2\sigma_1 \alpha_1 (1 - \bar{\alpha}_1)} \| \boldsymbol{\epsilon} - \boldsymbol{\epsilon}_{\theta, f_\phi(\mathbf{z}_1, t)}(\mathbf{x}_1, 1) \|^2 \Big] \Big] \\
&= C_1 + \mathbb{E}_{\boldsymbol{\eta}, \boldsymbol{\epsilon}, \mathbf{x}_0} \Big[ \frac{\beta_1^2}{2\sigma_1 \alpha_1 (1 - \bar{\alpha}_1)} \| \boldsymbol{\epsilon} - \boldsymbol{\epsilon}_{\theta, f_\phi(g_\xi(\cdot), t)}(\sqrt{\bar{\alpha}_1} \mathbf{x}_0 + \sqrt{1 - \bar{\alpha}_1} \boldsymbol{\epsilon}, 1) \|^2 \Big]
\end{aligned}
\tag{61}
$$

where $\eta, \epsilon \sim \mathcal{N}(\mathbf{0}, \mathbf{I})$, and the second equality is also derived by LOTUS.

Finally, by combining Eq. (58) and Eq. (61), we have

$$
\begin{aligned}
\mathbb{E}_q[-\log p_{\bar{\theta}}^{\mathrm{SMD}}(\mathbf{x}_0)] &\leqslant \mathcal{L}^{\mathrm{SMD}} = \sum_{t=0}^{T} \mathcal{L}_t^{\mathrm{SMD}} \\
&= C + \sum_{t=1}^{T} \mathbb{E}_{\boldsymbol{\eta}, \boldsymbol{\epsilon}, \mathbf{x}_0}\left[\Gamma_t \|\boldsymbol{\epsilon} - \boldsymbol{\epsilon}_{\theta, f_\phi(\cdot)}(\sqrt{\bar{\alpha}_t}\mathbf{x}_0 + \sqrt{1 - \bar{\alpha}_t}\boldsymbol{\epsilon}, t)\|^2\right]
\end{aligned}
,
\tag{62}
$$

where $C = \sum_{t=1}^{T} C_t$ and $\Gamma_t = \beta_t^2/(2\sigma_t \alpha_t (1 - \bar{\alpha}_t))$.

## G   GENERATED SAMPLES

Some images generated by our models (e.g., LDM w/ SMD) are in Fig. 5 and Fig. 6.

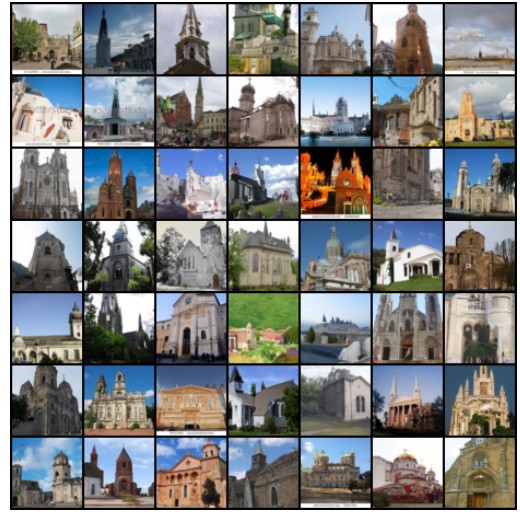 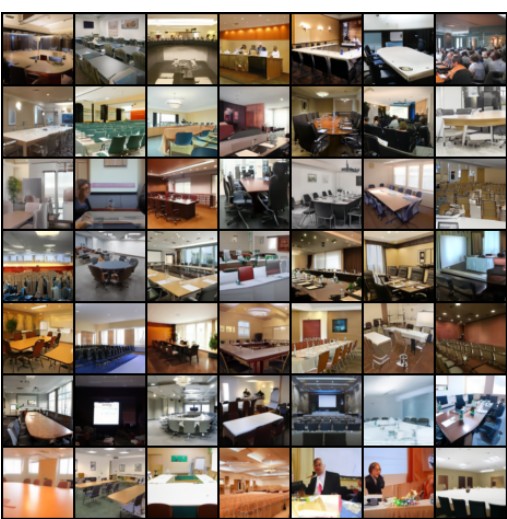

(a) Synthesized images of LSUN Church          (b) Synthesized images of LSUN Conference

Figure 5: $64 \times 64$ images generated by DDPM w/ SMD.

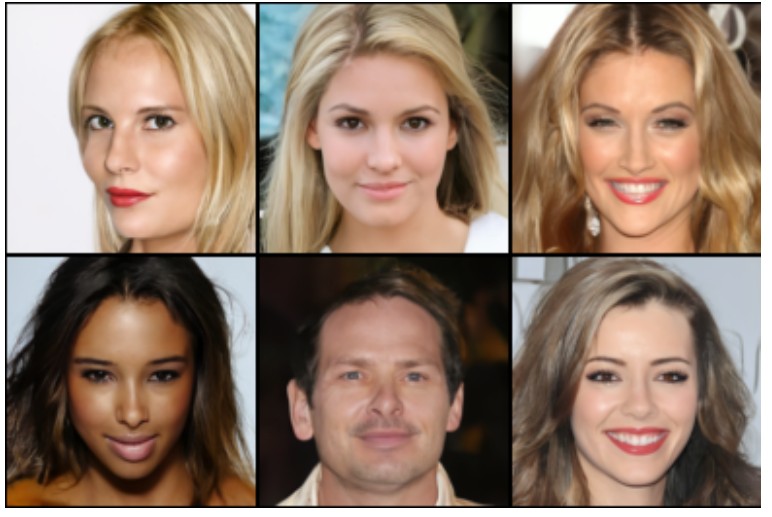 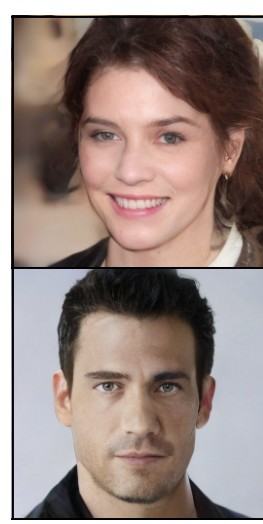

Figure 6: Generated images on CelebA-HQ $128 \times 128$ (left) and $256 \times 256$ (right). The left samples are from DDPM w/ SMD and the right ones from LDM w/ SMD.

