# OpenReview forum: "Soft Mixture Denoising: Beyond the Expressive Bottleneck of Diffusion Models"
_ICLR.cc/2024/Conference — ICLR 2024 poster_

### Official Review · Reviewer_nQBi · 2023-10-25

**Soundness:** 4 excellent
**Presentation:** 3 good
**Contribution:** 3 good
**Rating:** 8
**Confidence:** 4

**Summary:**

This paper...
- theoretically shows DMs suffer from an expressive bottleneck due to the assumption that the denoising distribution is Gaussian,
- proposes soft mixture denoising to address this problem,
- shows SMD improves the performance of DMs on various datasets.

**Strengths:**

- This paper proposes a novel approach to improving the sampling efficiency of diffusion models.
- SMD is well-motivated through rigorous theoretical analysis.
- This paper well-written and I had no trouble following the logic.
- There are non-trivial performance improvements after applying SMD.

**Weaknesses:**

- SMD requires training of additional $g_\xi$ and $f_\phi$ networks, so I would expect training SMD requires more VRAM and time compared to training standard diffusion models. A comparison of VRAM / training time / inference time of SMD vs. standard diffusion would be insightful.

**Questions:**

- Is SMD compatible with fast samplers such as EDM [1]? If it is, can the authors provide results? If not, can the authors suggest how SMD could be modified to be compatible with such fast samplers?
- How does the performance of SMD vary as we change the size of $g_\xi$ and $f_\phi$ networks? Does SMD work better if we use larger networks or is it sufficient to use small networks?

[1] Elucidating the Design Space of Diffusion-Based Generative Models, Karras et al., NeurIPS, 2022.

---

> ### Author Response · Authors · 2023-11-16
>
> Dear Reviewer nQBi,
>
> Thanks for your kind and constructive feedback.
>
> ### Comment 1
>
> "*… so I would expect training SMD requires more VRAM and time … A comparison of VRAM / training time / inference time of SMD vs. standard diffusion would be insightful.*"
>
> ### Response 1
> Thanks for raising this point. Since the components of our proposed SMD $g_{\xi}, f_{\phi}$ are much smaller than the backbone of diffusion models (i.e., U-Net), it has minor impact on the model size, training time, and inference speed. The following table is an empirical comparison between DDPM and DDPM w/ SMD on LSUN-Church 64x64.
>
> | Model         | GPU Memory (bsz=128) | Training Time Per Step | Inference Time (T=1000, bsz=49) |
> | ------------- | -------------------- | ----------------------- | -------------------------------- |
> | DDPM          | 35GB                 | 0.27s                  | 3min41s                          |
> | DDPM w/ SMD    | 38GB                 | 0.35s                  | 4min02s                          |
>
> where "bsz" denotes batch size. We see that SMD indeed causes minor increases in the GPU memory and running times.
>
> Notably, as seen in Figs. 2 and 3 of the original manuscript, SMD tends to require fewer training and inference steps, respectively, hence **SMD will in general require less overall training and inference time than vanilla DDPM**.
>
> ### Comment 2
> *"Is SMD compatible with fast samplers such as EDM [1]? If it is, can the authors provide results? If not, can the authors suggest how SMD could be modified to be compatible with such fast samplers?"*
>
> ### Response 2
> Yes, our proposed SMD is compatible with fast samplers---including the stochastic sampler of EDM [1], which applied Langevin dynamics to correct the predictions of denoising neural networks. We perform experiments on CIFAR-10 and LSUN-Conference to show how the EDM sampler and SDM interact. The results are as follows.
>
> | Model                           | CIFAR-10 | LSUN-Conference |
> | ------------------------------- | -------- | --------------- |
> DDPM | 3.78 | 4.15
> DDPM w/ EDM Sampler | 3.57 | 3.95
> | DDPM w/ SMD                     | 3.13     | 3.52            |
> | DDPM w/ SMD, EDM Sampler        | 2.95     | 3.21            |
>
>
> We observe that the stochastic sampler of EDM further improves DDPM and our model (i.e., DDPM w/ SMD) on both datasets. For example, the FID score of our model on LSUN-Conference is reduced by 8.81%. Perhaps more importantly, we see that SMD improves the FID for both datasets, with and without EDM. This indicates the advantage of SMD is separate from that of a specific sampler, and that a better sampler can still benefit from SMD.
>
> ### Comment 3
> *"How does the performance of SMD vary as we change the size of $g_{\xi}$ and $f_{\phi}$ network …"*
>
> ### Answer 3
> Thanks for raising this point. We built $g_{\xi}, f_{\phi}$ both as feedforward networks.  Empirically, we found that larger networks $g_{\xi}, f_{\phi}$ lead to better performances, but the marginal improvements decrease as the networks get bigger. Take $g_{\xi}$ with $128$ hidden dimensions as an example, the FID scores of our model (e.g., DDPM w/ SMD) for different network layers as follows:
>
> | Layers of $g_{\xi}$ | CIFAR-10 |
> | ------------------- | -------- |
> | 2                   | 3.25     |
> | **4 (main paper)**       | **3.13**     |
> | 6                   | 3.06     |
> | 8                   | 3.02     |
>
> We see that larger networks indeed result in better performance, but improvement increases become rather marginal.

---

> > ### Comment · Reviewer_nQBi · 2023-11-17
> >
> > Thank you for the reply! I'm satisfied with the authors' feedback, and will maintain my current score.

---

> > > ### Author Response · Authors · 2023-11-17
> > >
> > > Thank you, and thanks again for you time and constructive feedback!

---

### Official Review · Reviewer_mAPH · 2023-10-26

**Soundness:** 3 good
**Presentation:** 3 good
**Contribution:** 3 good
**Rating:** 8
**Confidence:** 4

**Summary:**

This paper studied the reverse process in the diffusion models. Specifically, the authors theoretically showed that the Gaussian assumption in the reverse process of the original diffusion models is not expressive enough for complicated target distribution, and proposed a soft-mixture model for the reverse denoising process. The authors theoretically demonstrated the expressiveness of the new model, and derived training and sampling algorithms for it. Experiments have been conducted to demonstrate the effectiveness of the proposed method.

**Strengths:**

1. The idea is new and reasonable.

2. The authors provided theoretical foundations for their proposed method.

3. The effectiveness of the proposed method has been empirically verified.

**Weaknesses:**

To me, there is no significant weakness of this work.

**Questions:**

To my knowledge, there are studies considers better parameterizing the distribution in the reverse process, such as:

1. Zhisheng Xiao, Karsten Kreis, and Arash Vahdat. Tackling the Generative Learning Trilemma with Denoising Diffusion GANs. ICLR, 2022.
2. Yanwu Xu, Mingming Gong, Shaoan Xie, Wei Wei, Matthias Grundmann, Kayhan Batmanghelich, and Tingbo Hou. Semi-Implicit Denoising Diffusion Models (SIDDMs). arXiv:2306.12511, 2023.

The authors should discuss these studies, and better to empirically compare with them.

---

> ### Author Response · Authors · 2023-11-16
>
> Dear Reviewer mAPH,
>
> Thanks for your kind and insightful feedback.
>
> ### Comment 1
> "*… there are studies considers better parameterizing the distribution in the reverse process … The authors should discuss these studies …*"
>
> ### Response 1
> Thank you for sharing these interesting papers. The first work proposed a model called DDGANs, which applied a conditional GAN for denoising a few-step diffusion process. The second paper (i.e., SIDDMs) further improved DDGANs by decomposing the adversarial loss, achieving better performance. Similar to our proposed SMD, GAN-based denoising models can also learn non-Gaussian denoising distributions. However, GAN is notoriously unstable for training and more sensitive to hyperparameter selection [1,2], while pure diffusion models are generally more stable.
>
> We will cite these papers and include the above discussion in the revised related work.
>
> ### References
>
> [1] Arjovsky and Bottou, Towards Principled Methods for Training Generative Adversarial Networks, ICLR-2017.
>
> [2] Lucic et al., Are GANs Created Equal? A Large-Scale Study, NeurIPS-2018.

---

> > ### Comment · Reviewer_mAPH · 2023-11-19
> >
> > Thanks for your reply. I will keep my initial rating.

---

> > > ### Author Response · Authors · 2023-11-19
> > >
> > > Thank you and thanks again for your efforts in reviewing our paper!

---

### Official Review · Reviewer_eLFe · 2023-10-31

**Soundness:** 3 good
**Presentation:** 3 good
**Contribution:** 3 good
**Rating:** 5
**Confidence:** 4

**Summary:**

this submission introduces soft mixture denoising for improving the expressive bottleneck of diffusion models. It first shows that diffusion models have an expressive bottleneck in the backward denoising steps, when approximating p(xt-1|xt) using a Gaussian distribution, that leads to unbounded local and global denoising. It then proposes soft mixture denoising (SMD) that approximate the backward step p(xt-1|xt) using a  gaussian mixture distribution, where the number of modes is infinity. This soft gaussian mixture is a universal approximator for continuous probability distributions and the result shows that the local and global errors would be bounded. Experiments with image datasets indicate that SMD improves different diffusion models such as DDPM and DDIM.

**Strengths:**

Improving the design of diffusion models and make them more efficient is a timely problem

Identifying the expressiveness bottleneck of single gaussian approximation, and the unbounded denoising errors is novel for denoising diffusion models

The experiments are extensive

**Weaknesses:**

The performance gains for the new soft mixture models are not significant. One would expect a significant reduction in number of steps if soft mixture is a better approximation for p(xt-1|xt), but that is not the case in the experiments.

The architectural changes for the new denoising networks are not discussed well. It’s a bit confusing how the

**Questions:**

The mean parameterization in eq. 11 needs to be clarified? What is the hyper network representation? what is \theta \cup f_{\phi}?

While the theory supports that soft gaussian mixture to be a universal approximator. However, the performance gains compared with single Gaussian are not significant. What are the limitations and approximations that led to that? Could it be the identity assumption for the covariance matrix?

---

> ### Author Response · Authors · 2023-11-16
>
> Dear Reviewer eLFe,
>
> We thank you for your kind and constructive feedback. We would like to respond to your questions in a point-by-point manner. Most importantly, **as shown in Fig. 1 and Fig. 3 of our paper, we highlight that our proposed SMD has significantly improved the diffusion models in the case of few denoising iterations. For example, SMD nearly doubly reduced the FID score of LDM on CelebA-HQ**.
>
> ### Comment 1
> "*The performance gains for the new soft mixture models are not significant. One would expect a significant reduction in number of steps if soft mixture is a better approximation for p(xt-1|xt), but that is not the case in the experiments.*"
>
> ### Response 1
> We would like to highlight that our proposed SMD approach is more effective for few backward iterations (e.g., $T= 100 < 1000$). **Fig. 3 of our paper shows a detailed comparison between DDPM and DDPW w/ SMD in terms of different backward iterations**. We extracted part of the results (on CelebA-HQ) from Fig. 3 and put them in the following table.
>
> Model                  |  T=100 |   T=200  |  T=600 |   T=800  |   T=1000
> --- | --- | --- | --- | --- | ---
> LDM               |         11.29   |     9.16    |  7.11      |   6.59   |    6.13
> LDM w/ SMD      |     6.85     |   6.33   |    5.87   |     5.51  |       5.48
>
> From these results, we conclude the following:
> 1) **SMD significantly improves vanilla diffusion models for few backward iterations**. For example, the FID score at $T=100$ is almost halved from $11.29$ to $6.85$;
> 2) **SMD improves the inference speed**. As a result of the above, we see that LDM w/ SMD is less sensitive to the number of steps and can achieve the same FID for fewer steps. For example, LDM w/ SMD (T=200) outperforms vanilla LDM (T=800). Fewer inference steps while retaining quality could additionally lower computational costs and promote wider model accessibility.
>
> In a nutshell, SMD does make significant improvements to existing diffusion models, enabling lower computational costs while retaining quality. We further highlighted this in the results section.
>
> ### Comment 2
> "*While the theory supports that soft gaussian mixture to be a universal approximator. However, the performance gains compared with single Gaussian are not significant. What are the limitations and approximations that led to that? Could it be the identity assumption for the covariance matrix?…*"
>
> ### Response 2
> We believe the performance gains of SMD are significant compared to the vanilla (single gaussian) approach—see our Response 1. For example, **SMD reduced the FID score of LDM on CelebA-HQ by half for $T=100$**, and improved vanilla diffusion models for full denoising iterations $T=1000$---see Table 2 of our paper, e.g. **SMD reduced the FID score of LDM on LSUN-Church 256x256 by $11.09\%$**.
>
> **Limitations**. Like standard diffusion models, SMD is trained with an upper bound of negative log-likelihood and the optimization of neural networks is non-convex. Consequently, in practice, we will observe some errors, even though a Gaussian mixture is a universal approximator. Additionally, you mentioned that the “identity assumption for the covariance matrix” may also have a negative impact. However, a diagonal covariance matrix is assumed in all diffusion model literature, as it would be intractable to model full covariance matrices in high dimensions. To conclude, we agree these are limitations, but for diffusion models in general.
>
> ### Comment 3
> "*The architectural changes for the new denoising networks are not discussed well. It’s a bit confusing how the [sic, no sentence ending]. The mean parameterization in eq. 11 needs to be clarified? What is the hyper network representation? what is $\theta \cup f_{\phi}$?*"
>
> ### Response 3
>
>
> Thank you for highlighting these unclarities. The architecture of our model and the details of Eq. (11) are as follows. Our model contains a variant of U-Net and parameterized modules $f_{\phi}, g_{\xi}$. In Eq. (11), both $\theta$ and $f_{\phi}(z_t, t)$ are the parameters of U-Net for computations. While $\theta$ represents typical parameters of the standard U-Net, $f_\phi$ is a hypernetwork that depends on latent variable $z_t$. In the implementation, we built $f_{\phi}$ as a FNN (i.e., feedforward neural network) that takes $(z_t, t)$ as the input and outputs the parameters of two FNNs that we place before and after the standard U-Net. For $g_{\xi}$, it is also a simple FFN that takes Gaussian noise and $(x_t, t)$ as the input and outputs latent variable $z_t$. Compared with regular diffusion models, our model has new modules $f_{\phi}, g_{\xi}$ and adds extra layers (that are dynamically computed from $f_{\phi}$) into a common U-Net.
>
> We will include these details in the revised manuscript.

---

> ### Author Response · Authors · 2023-11-20
>
> Dear Reviewer eLFe,
>
> We thank you for your time in reviewing our paper. With only 2 days left, we would like to know whether our previous response has addressed your concerns. Looking forward to your feedback!
>
> Best regards,
>
> The Authors

---

> ### Author Response · Authors · 2023-11-22
>
> Dear Reviewer eLFe,
>
> As we have not heard from you and the discussion period ends within a day, we would like to confirm with you whether our rebuttal has addressed all your concerns. If not, please let us.
>
> Many thanks!
>
> Authors of #4096

---

### Official Review · Reviewer_izts · 2023-11-05

**Soundness:** 2 fair
**Presentation:** 3 good
**Contribution:** 2 fair
**Rating:** 6
**Confidence:** 4

**Summary:**

This paper identifies an expressive bottleneck in the backward denoising process of current diffusion models, challenging the strong assumptions underlying their theoretical guarantees. The authors demonstrate that these models can incur unbounded errors in both local and global denoising tasks. To address this, they introduce Soft Mixture Denoising (SMD), a more expressive model that theoretically can approximate any Gaussian mixture distribution. The effectiveness of SMD is validated through experiments on various image datasets, particularly noting significant improvements in diffusion models, like DDPM, with few backward iterations.

**Strengths:**

1. The paper is articulate and presents a clear logical progression.

2. Both theoretical exposition and experimental verification are provided to substantiate the authors’ arguments.

**Weaknesses:**

1. The critique leveled against existing diffusion models seems to be somewhat overstated. These models have achieved considerable success across various applications and can represent complex distributions effectively. The alleged expressive bottleneck is contingent upon the noise scheduling strategy deployed. For instance, in typical diffusion models, a small value of $\beta_t$, such as 0.0001, is assumed to be used as the initial. As indicated in Equation (25), the transition probability $q(x_{t-1} | x_t)$ approaches a Gaussian distribution as $\beta_t$ tends toward zero, which contradicts the claim of an inherent expressive limitation.

2. The selection of datasets for experimentation—LSUN and CelebA—seems narrow given the criticism of diffusion models' multimodality capacity. For a robust evaluation, a more complex and varied dataset like ImageNet, encompassing 1k categories, would be more appropriate.

3. There appears to be a flaw in the derivation of local denoising error $M_t$. The associated loss term in $L_{t-1}$ is predicated on the KL divergence $KL[q(x_{t-1} | x_t, x_0) || p_\theta(x_{t-1} | x_t)]$. Here, $q(x_{t-1} | x_t, x_0)$, which is a known Gaussian distribution, should not be conflated with $q(x_{t-1} | x_t)$, which represents an unknown distribution. The validity of Theorems 3.1 and 3.2 is reliant on the accurate definition of $M_t$.

4. The paper does not reference the FID (Fréchet Inception Distance) results from the Latent Diffusion Model (LDM) study. In the LDM research, the reported FID scores were 4.02 for LSUN-Church and 5.11 for CelebA-HQ, which are superior to the performance metrics achieved by SMD as presented in this paper. This omission is significant as it pertains to the comparative effectiveness of the proposed model.

**Questions:**

1. There seems to be a typographical error involving a comma in the superscript at the end of Equation (3).
2. Could you detail the noise schedule utilized in your algorithm? The experiments section suggests that the original DDPM scheduling is retained while only the model component is modified. Considering that your paper emphasizes the importance of shorter chains and the expressiveness issue within them, it would be beneficial to see experimentation with significantly fewer steps to underscore the advantages of your proposed Soft Mixture Denoising (SMD).
3. The SMD approach bears resemblance to a Variational Autoencoder (VAE) in its structure. Could you confirm if this observation is accurate or elaborate on the distinctions between SMD and VAE?

---

> ### Author Response · Authors · 2023-11-16
> **Part-1 of Our Response**
>
> Dear Reviewer izts,
>
> Thank you for your constructive and comprehensive feedback. In the following, we aim to answer your concerns in a point-by-point manner. Most importantly, we (i) show that our defined error $M_t$ is valid, (ii) perform new experiments on ImageNet, and (iii) compare our model with LDM using their paper's experimental settings.
>
> ***We begin by answering your 3rd comment (on the validity of the definition of $M_t$), as our other responses rely on this.***
>
> ### Comment 3
> "*There appears to be a flaw in the derivation of local denoising error $M_t$. The associated loss term in $L_{t-1}$ is predicated on the KL divergence... Here, $q(x_{t_1} | x_t, x_0)$ which is a known Gaussian distribution, should not be conflated with $q(x_{t-1} | x_t)$, which represents an unknown distribution. The validity of Theorems 3.1 and 3.2 is reliant on the accurate definition of $M_t$.*"
>
> ### Response 3
> Thank you for raising this concern. We suspect that your concern stems from Eq. (3) of our paper, which includes the term $L_{t-1}$  *but also other terms*. When combined with the other terms, the full DDPM loss is $L_{p_{\theta}} = E_{q}[-\ln \frac{p_{\theta}(x_{0:T})}{q(x_{1:T}\vert x_0)}]$ (see also Eq. 3 of [1]). With the following proposition, we will show that $M_t$ as defined in the paper (with $q(x_{t-1} | x_t)$ and **not** $q(x_{t-1} | x_t, x_0)$) is appropriate for studying this loss.
>
> **Proposition: For a perfectly optimized diffusion model $p_{\theta} = \arg\min_{p_{\theta}'} L_{p_{\theta}'}$, the equality $p_{\theta}(x_{t-1} | x_t) = q(x_{t-1} | x_t)$ holds for every denoising iteration $t \in [1, T]$. Here the two terms $p_{\theta}, L_{p_{\theta}}$ respectively represent a diffusion model $[ p_{\theta}(x_{t-1} | x_t) ]_{t \in [1, T]}$ and its loss**.
>
> **Proof**: Following DDPM [1], we formulate the loss function of diffusion models as
>
> $L_{p_{\theta}} = E_{q}[-\ln \frac{p_{\theta}(x_{0:T})}{q(x_{1:T}|x_0)}] = D_{KL}(q(x_{0:T}) || p_{\theta}(x_{0:T})) - E_q[q(x_0)]$.
>
> Note that the second expectation term $E_q[\cdot]$ is a constant and the first KL-divergence term $D_{KL}(\cdot)$ reaches its minimum $0$ when $p_{\theta}(x_{0:T})$ equals $q(x_{0:T})$. Therefore, for a perfectly optimized diffusion model $p_{\theta} = \min_{p_{\theta'}} L_{p_{\theta'}}$, we have $p_{\theta}(x_{0:T}) = q(x_{0:T})$. Then, for every iteration $t \in [1, T]$, we have
>
> $p_{\theta}(x_t, x_{t-1}) = \int p_{\theta}(x_{0:T}) dx_{1:t-2} dx_{t+1:T} = \int q(x_{0:T}) dx_{1:t-2} dx_{t+1:T} = q(x_t, x_{t-1})$.
>
> Similarly, we get $p_{\theta}(x_t) = \int p_{\theta}(x_t, x_{t-1}) dx_{t,t-1} = \int q(x_t, x_{t-1}) dx_{t,t-1} = q(x_t)$. Based on the above two equations, we finally derive
>
> $p_{\theta}( x_{t-1} \vert x_t) = \frac{p_{\theta}( x_{t-1}, x_t)}{p_{\theta}(x_t)} = \frac{q( x_{t-1}, x_t)}{q(x_t)} = q( x_{t-1} \vert x_t)$,
>
> proving the proposition.
>
> **From the above proposition, we can see that $p_{\theta}(x_{t-1} | x_t)$ is in fact optimized towards $q(x_{t-1} | x_t)$ for minimizing $L_{p_{\theta}}$. Therefore, it is appropriate to define the denoising error $M_t$ of $p_{\theta}(x_{t-1} | x_t)$ as its distributional gap to $q(x_{t-1} | x_t)$, and not $q(x_{t-1} | x_t, x_0)$**.

---

> ### Author Response · Authors · 2023-11-16
> **Part-2 of Our Response**
>
> ### Comment 1
> "*The critique leveled against existing diffusion models seems to be somewhat overstated. These models have achieved considerable success across various applications … The alleged expressive bottleneck is contingent upon the noise scheduling strategy … the transition probability approaches a Gaussian distribution as $\beta_t$ tends toward zero …*"
>
> ### Response 1
> Diffusion models are indeed performing well and have many successful applications (e.g., Midjourney). We also agree that a mixture model becomes unnecessary when $T\rightarrow \infty$. In practice, however, large $T$ requires more computations and slows down sampling. Thus, current diffusion models are not perfect. For example, **recent works (e.g., DDIM [2] and DPM [3]) show that their performances degrade significantly for few backward iterations**. Our paper aims to solve this problem by introducing a new backward denoising paradigm that makes diffusion models more expressive. **As shown in our paper (Fig. 1 and Fig.3), diffusion models with the proposed SMD approach are much less affected by the number of backward iterations**. Consequently, we believe our proposed SMD has addressed a serious weakness of current diffusion models and can improve sampling speed and reduce computational cost.
>
> Regarding your concerns about “expressive bottleneck being contingent upon the noise scheduling strategy” and $\beta_t$, we note that **our Theorem 3.1, which shows the denoising error $M_t$ is uniformly unbounded, is independent of the variance schedule $\beta_t$. In other words, regardless of the scale of $\beta_t$, $q(x_{t-1} | x_t)$ can be arbitrarily too complex for $p_{\theta}(x_{t-1} | x_t)$ to approximate**. Additionally, $\beta_t$ is only initially small and gradually becomes non-negligible as $t$ increases from $1$ to $T$. As evidence, Fig. 1 and Fig. 3 of our paper show that vanilla diffusion models perform poorly with few backward denoising iterations, where $\beta_t$ becomes large even for small $t$.
>
> ### Comment 2
> "*… For a robust evaluation, a more complex and varied dataset like ImageNet, encompassing 1k categories, would be more appropriate.*"
>
> ### Response 2
> Thanks for pointing this out. Based on your suggestion, **we have conducted new experiments on ImageNet** and the results are summarized below.
>
> | Model       | FID scores on ImageNet 64x64 |
> |-------------|------------------------------|
> | DDPM        | 3.76                         |
> | **DDPM w/ SMD** | **2.87**                         |
> | ADM         | 2.13                         |
> | **ADM w/ SMD**  | **1.65**                         |
>
> From the above table, we can see that our proposed SMD significantly improves both DDPM and ADM on ImageNet. For example, SMD reduces the FID score of DDPM by nearly 1 point. **The results indicate that SMD is applicable to more complex and varied datasets**. We thank the reviewer for making this point which further helped us highlight the merits of our approach.
>
> ### Comment 4
> "*… In the LDM research, the reported FID scores were 4.02 for LSUN-Church and 5.11 for CelebA-HQ, which are superior to the performance metrics achieved by SMD as presented in this paper …*"
>
> ### Response 4
> Thank you for raising this point. Our reported FID scores cannot be directly compared to the original paper, since **our model and LDM [4] are not evaluated in the same experimental set-up**. For example, LDM uses more data and computing resources to train the separate VAE and they adopted different hyperparameters (e.g., number of U-Net layers).
>
> **To allow for a fairer comparison, we have examined the original LDM paper [4] and followed most of its settings**. For example, (i) hidden dimensions are respectively 224 and 192 for CelebA-HQ and LSUN-Church, (ii) channel multipliers are {1,2,3,4} for CelebA-HQ and {1,2,2,4,4} for LSUN-Church, (iii) models are trained with 410K iterations on CelebA-HQ and 500K on LSUN-Church, and (iv) DDIM is applied for sampling, with 500 iterations on CelebA-HQ and 200 iterations on LSUN-Church. **Below are the new experimental results**:
>
> Model             |      Result Source  |           LSUN-Church   |       CelebA-HQ
> --- | --- | --- | ---
> LDM |                         LDM [4] |                       4.02     |                 5.11
> LDM |                       Our Experiment |             4.15          |               5.27
> **LDM w/ SMD**  |         **Our Experiment**  |           **3.71**                 |        **4.52**
>
> From the above table, we can see that:
> 1) Our reported  FID scores of LDM are consistent with the results of its original paper [4];
> 2) **Our model (i.e., LDM w/ SMD) significantly outperforms LDM in terms of both our reported results and those from its original paper [4]**. For example, the FID score on CelebA-HQ is reduced by 11.54%.
>
> We thank the reviewer again for pointing out this discrepancy, which allows us to shed new light on our method.

---

> ### Author Response · Authors · 2023-11-16
> **Part-3 of Our Response**
>
> ### Other questions
>
> **Question 1**: There seems to be a typographical error involving a comma in the superscript at the end of Equation (3).
>
> **Response Q1**: We have now removed this. Thanks!
>
> **Question 2**: Could you detail the noise schedule utilized in your algorithm … it would be beneficial to see experimentation with significantly fewer steps to underscore the advantages of your proposed Soft Mixture Denoising (SMD).
>
> **Response Q2**: For the noise schedule, we have followed the linear schedule of the standard DDPM [1], increasing from $\beta_1 = 10^{-4}$ to $\beta_T = 0.02$. For your suggested “experimentation with significantly fewer steps”, please check **Fig. 1 and Fig. 3 of the paper, which both show our proposed approach significantly improves current diffusion models in the case of few backward iterations**. For example, Fig. 1 shows that the diffusion model with our approach achieves an FID score of $6.85$ with only $T = 100$ backward iterations, which is almost half of the FID performance (i.e., $11.29$) of the vanilla diffusion model with the same number of backward iterations.
>
> **Question 3**: The SMD approach bears resemblance to a Variational Autoencoder (VAE) in its structure. Could you confirm if this observation is accurate or elaborate on the distinctions between SMD and VAE?
>
> **Response Q3**: Our proposed approach and VAE both have a concept of latent variables, but they are very different. While VAE builds mappings between real samples and latent variables, SMD resorts to a latent variable $z_t$ to only capture the potential mixture structure $q(z_t | x_t)$ of $q(x_{t-1} | x_{t})$. Therefore, in contrast to VAEs, the SMD $z_t$ does not encode a full sample, nor do we put a prior on its distribution (cf. the usually Gaussian prior used in VAEs).
>
> ### References
>
> [1] Ho et al., Denoising Diffusion Probabilistic Models, NeurlPS-2020.
>
> [2] Song et al., Denoising Diffusion Implicit Models, ICLR-2021.
>
> [3] Lu et al., DPM-Solver: A Fast ODE Solver for Diffusion Probabilistic Model Sampling in Around 10 Steps, NeurlPS-2022.
>
> [4] Rombach et al., High-Resolution Image Synthesis with Latent Diffusion Models, CVPR-2022.

---

> ### Comment · Reviewer_izts · 2023-11-17
> **Response**
>
> Thank you for including the ImageNet results and the new comparison with the LDM in your revision. These additions certainly contribute to the robustness of your paper.
>
> Upon reviewing your response, I acknowledge the efforts to address previous concerns. However, I maintain a critical concern regarding the efficiency of the proposed model.
>
> In your response, you outline the aim of your paper: to introduce a novel backward denoising paradigm to enhance the expressiveness of diffusion models, thereby reducing the number of backward steps and improving sampling speed. The central claim is that when the diffusion chain is truncated, the posterior $q(x_{t-1} | x_t)$ deviates from Gaussianity, hence your proposal of employing a Gaussian mixture-based model SMD as a replacement for the standard Gaussian model.
>
> While the observation about non-Gaussian posteriors in shortened diffusion chains is valid, this has been noted and studied in prior works (e.g., Figure 2 of DDGAN [1]). DDGAN's approach, using a GAN objective to address multimodality in diffusion steps, demonstrated the ability to train with just 4 steps and sample within the same. In contrast, your experiments still require approximately 1000 steps for training and 100 steps for sampling. If the paper's key contribution lies in efficiency, then it appears there is a significant gap when compared to DDGAN's achievements, considering that both models aim to tackle multimodality in a reduced number of diffusion steps.
>
> While I recognize the strides made in your approach to shorten the backward steps in diffusion models, it is important to benchmark this improvement against contemporary models that set a higher standard in efficiency. Current consistency models have demonstrated the capability to generate samples in as few as 1 to 2 steps. Furthermore, leveraging advanced Ordinary Differential Equation (ODE) solvers such as DPM or EDM allows for sampling to be compressed into a mere 10 steps. In this context, the 100 steps for sampling as presented in your experiments do not align with the cutting-edge advancements in the field. It would be beneficial for the paper to address these developments and re-evaluate the proposed model's efficiency in light of these high-performing techniques.
>
> Additionally, I suggest a revision of the initial sentence in the "derivation of local denoising error" paragraph. The sentence should be clarified to ensure that the definition of the loss term $L_{t-1}$ is accurately reflected and not contingent on an imprecise interpretation.
>
> Reference:
> [1] Xiao, Z., Kreis, K., & Vahdat, A. (2021). Tackling the generative learning trilemma with denoising diffusion GANs. arXiv preprint arXiv:2112.07804.

---

> ### Author Response · Authors · 2023-11-18
>
> Dear Reviewer izts,
>
> We are glad that you are satisfied with our previous answers and thank you for the new feedback.
>
> ## Comparisons with DDGAN and Consistency Models
>
> Thanks for mentioning these two models: DDGAN and Consistency Models.
>
> As we answered to Reviewer mAPH, **DDGAN adopts GAN as the backbone, but GANs are known for unstable training [1] and have high sensitivity to hyperparameter selection [2]**. In contrast, pure diffusion models are generally more stable while achieving high generation quality. To provide a comparison with DDGAN, we followed DDGAN’s paper and **conducted a new experiment with very few backward iterations**. The results are as follows.
>
> | Model                        | Result Source               | Backward Iterations | FID Score on CIFAR-10 |
> | ---------------------------- | --------------------------- | ---------- | ---------------------- |
> | DDGAN                        | Table 1 in DDGAN [3]        | T=4        | 3.75                 |
> | DDGAN                        | Table 2 in DDGAN [3]        | T=8        | 4.36                 |
> | Our Model: DDPM w/ SMD        | Our experiment             | T=4        | 3.83                 |
> | Our Model: DDPM w/ SMD        | Our experiment             | T=8        | **3.69**                 |
> | Our Model: DDPM w/ SMD        | Our experiment             | T=16       | **3.60**                 |
>
> We draw the following two conclusions:
> 1) Our model performs closely to DDGAN for $T=4$ and achieves better performances for $T=8$. **Therefore, the proposed SMD is also applicable to the case of extremely few denoising iterations**;
> 2) Unlike our model that performs better with increasing backward iterations $T$, the FID score of DDGAN does not improve beyond $T=4$---surprisingly, performance goes down significantly for $T=8$. This indicates **DDGAN might not be scalable to more backward iterations**, making it less applicable to tasks that expect very high generation quality.
>
> Consistency Models (CMs) are indeed promising for very few-step diffusion. However, **it is with knowledge distillation that CMs performed comparably with other diffusion models, and otherwise perform less well**. For example, the CMs paper reported a FID score of $2.93$ on CIFAR-10 for $2$ backward iterations, but without knowledge distillation, the score was significantly reduced to $5.83$.
>
> ## Minor Points
>
> Comment 1: DPM or EDM allows for sampling to be compressed into a mere 10 steps. In this context, the 100 steps for sampling as presented in your experiments do not align with the cutting-edge advancements in the field.
>
> Answer 1: Thanks for raising this point. For DPM, its original paper [4] reported that it achieved an FID score of $4.70$ on CIFAR-10 with $10$ backward iterations, which is significantly larger than that of our model (i.e., $3.69$) with fewer iterations $T=8$ (see the above table). For EDM, its sampler is compatible with our proposed SMD for application: please refer to our Response 2 to Reviewer nQBi.
>
>
> Comment 2: The sentence should be clarified to ensure that the definition of the loss term is accurately reflected
>
> Answer 2:  Thanks for your suggestion. We will revise that sentence to “The term $L_{t-1}$ in Eq. (3) indicates that it is appropriate to apply KL-divergence to measure the distributional discrepancy. In the appendix, we will prove that $p_{\theta}(x_{t-1} | x_t)$ is optimized towards $q(x_{t-1} |  x_t)$. Therefore, we define the error $M_t$ as the KL-divergence between the two distributions.”
>
> ## References
>
> [1] Arjovsky and Bottou, Towards Principled Methods for Training Generative Adversarial Networks, ICLR-2017.
>
> [2] Lucic et al., Are GANs Created Equal? A Large-Scale Study, NeurIPS-2018.
>
> [3] Xiao et al., Tackling the Generative Learning Trilemma with Denoising Diffusion GANs, ICLR-2022.
>
> [4] Lu et al., DPM-Solver: A Fast ODE Solver for Diffusion Probabilistic Model Sampling in Around 10 Steps, Neurips-2022.

---

> > ### Comment · Reviewer_izts · 2023-11-20
> > **Response**
> >
> > I appreciate the authors' efforts during the rebuttal phase. The conducted comparison experiments with DDGAN (T=4), CM (T=2), and EDM (T=18) are crucial for demonstrating the efficiency improvement in diffusion models. However, presenting SMD with a sampling capability of T=100 steps is not particularly impressive or groundbreaking compared to these advanced techniques. While I would have liked to see these experiments applied to a broader range of datasets, I understand that such an extension might be infeasible within the constraints of the rebuttal phase. Nevertheless, I have increased my score in recognition of the significant work the authors have put into the rebuttal.
> >
> > Regarding the application of Gaussian Mixture to learn a shorter reverse process, I find this approach to be a straightforward and viable alternative to GAN-based methods. The experimental results seem to confirm my initial thoughts. However, I did not observe any distinctly superior aspects of using Gaussian Mixture compared to GAN, CM, and EDM methods. While exploring Gaussian Mixture is a worthwhile endeavor, it doesn't strike me as particularly promising at this moment.
> >
> > Additional Information: On CIFAR-10, the improved version of CM can achieve an FID of approximately 2.2 with two steps and 2.5 with one step. Meanwhile, EDM reaches an FID of around 2 with 18 steps.
> >
> > Additional Note: The SMD model performs well with T=4 steps as shown in the newest experiment why the original paper tested it on T=1000 steps?

---

> > > ### Author Response · Authors · 2023-11-20
> > >
> > > We aimed to show that our proposed SMD improved vanilla diffusion models for both many (Table 1 of our paper) and few (Fig. 3) backward iterations. As you suggested, we will add the new results of our model in the case of extremely few denoising iterations (e.g., $T=4$) in the revised version. We also thank you for the constructive reviews and your participation during this period!

---

### Meta-Review · Area_Chair_TQmk · 2023-12-05

**Metareview:**

The paper proposes soft mixture denoising where the Gaussian noise distribution in the diffusion process is replaced with a soft mixture. Using theoretical and empirical arguments, the authors show that the mixture model is more expressive and can lead to better results.

As pointed by reviewer izts , the critique leveled against existing diffusion models seems to be somewhat overstated. Diffusion models with gaussian distribution has been hugely successful. So, challenging this assumption needs very strong concrete experiments to validate.

One of the main concerns raised by reviewers was the lack of experimental results in the initial version. In the rebuttal, the authors have included some important results - results on Imagenet, comparison with DDGAN which also includes results on low sampling steps (8 steps). With these results, the quality of the paper has significantly improved. I agree with reviewer izts that presenting SMD with a sampling capability of T=100 steps is not particularly impressive or groundbreaking. However, showing comparison on less steps is what makes it appealing. The experiments included in the rebuttal proves this point to some extent and the authors should include this in the paper.

So, I am currently leaning towards acceptance, but I am really on the fence because this is a lot of new experiments added. The narrative of the experiment section should be changed accordingly in the revised draft. The current paper draft without experiments in the rebuttal is not good enough.

**Justification For Why Not Higher Score:**

The experiments are not too strong. To make it to the level of an oral paper, the paper needs to have significant comparisons and good analysis for why mixture distribution is better. Large scale experiments are missing too.

**Justification For Why Not Lower Score:**

The paper addresses a fundamental problem in the diffusion models, and the results included in the rebuttal (Imagenet, comparison with DDGAN) are quite good.

---

### Decision · Program_Chairs · 2024-01-16

Accept (poster)